# Drivers of Laptev Sea interannual variability in salinity and temperature

Phoebe A. Hudson[1, 2], Adrien C. H. Martin[2, 3], Simon A. Josey[2], Alice Marzocchi[2], Athanasios Angeloudis[1]

[1] University of Edinburgh, Edinburgh UK

[2] National Oceanography Center, Southampton, UK

[3] NOVELTIS, Labège, France

*Correspondence to*: Phoebe A Hudson (PA.Hudson@sms.ed.ac.uk)

**Abstract.** Eurasian Rivers provide a quarter of total fresh water to the Arctic, maintaining a persistent fresh layer that covers the surface Arctic Ocean. This freshwater export controls Arctic Ocean stratification, circulation, and basin-wide sea ice concentration. The Lena River supplies the largest volume of runoff and plays a key role in this system, as runoff outflows into the Laptev Sea as a particularly shallow plume. Previous in-situ and modelling studies suggest that local wind forcing is a driver of variability in Laptev sea surface salinity (SSS) but there is no consensus on the roles of Lena River discharge and sea ice cover in contributing to this variability or on the dominant driver of variability. Until recently, satellite SSS retrievals were insufficiently accurate for use in the Arctic. However, retreating sea ice cover and continuous progress in satellite product development have significantly improved SSS retrievals, giving satellite SSS data true potential in the Arctic. In this region, satellite-based SSS is found to agree well with in-situ data (r > 0.8) and provides notable improvements compared to the reanalysis product used in this study (r > 0.7) in capturing patterns and variability observed in in-situ data.

This study demonstrates a novel method of identifying the dominant drivers of interannual variability in Laptev Sea dynamics within reanalysis products and testing if these relationships appear to hold in satellite-based SSS, sea surface temperature (SST) data and in-situ observations. The satellite SSS data firmly establishes what is suggested by reanalysis products and what has previously been subject to debate due to the limited years and locations analysed with in-situ data; the zonal wind is the dominant driver of offshore or onshore Lena River plume transport. The eastward wind confines the plume to the southern Laptev Sea and drives alongshore transport into the East Siberian Sea and westward wind drives offshore plume transport into the northern Laptev Sea. This finding is affirmed by the strong agreement in SSS pattern under eastward and westward wind regimes in all reanalyses and satellite products used in this study, as well as with in-situ data. The pattern of SST also varies with the zonal wind component and drives spatial variability in sea ice concentration.

**Key Points:**

- The zonal wind component is the dominant driver of Lena River plume transport, with strong patterns in satellite and reanalysis-based sea surface salinity (SSS), sea surface temperature (SST) and sea ice concentration (SIC) products.
- The eastward wind confines the plume to the southern Laptev Sea and drives alongshore transport into the East Siberian Sea and westward wind drives offshore plume transport towards the northern Laptev Sea.
- There is no evidence that cumulative spring, summer or annual Lena River runoff plays a notable role in controlling interannual surface plume transport.

**1 Introduction:**

Dramatically warming Arctic surface air temperatures have altered Arctic atmospheric circulation and caused ocean warming, an intensification of the hydrological cycle, snow and ice melt, and increases in river runoff (Overland and Wang, 2010; Prowse et al., 2015). These changes have the potential to drive enhanced stratification with increases in freshwater input (in the form of runoff and precipitation), or increased mixing (with the loss of sea ice and resulting increasing atmosphere-ocean heat and momentum transfer) (IPCC (Intergovernmental Panel on Climate Change), 2019). Understanding the interplay between these changes is crucial for predicting the future state of the Arctic system.

The Laptev Sea, within the Eurasian Arctic (Figure 1), provides an ideal region to study the interactions between these changes, as a hotspot of Arctic warming, sea ice loss, and increases in river runoff (Kraineva and Golubeva, 2022; Stadnyk et al., 2021). Changes in this region will likely have considerable influence on the wider Arctic as the Laptev Sea is a key region of Arctic sea ice production and dominant contributor to Arctic-wide thermohaline structure, including to the surface Transpolar Drift and to the Beaufort Gyre (Johnson and Polyakov, 2001; Morison et al., 2012; Reimnitz et al., 1994; Thibodeau et al., 2014). The combination of these changes will also have considerable local impacts, including by increasing coastal erosion, altering nutrient availability and primary productivity (Juhls et al., 2020; Nielsen et al., 2020; Paffrath et al., 2021; Polyakova et al., 2021).

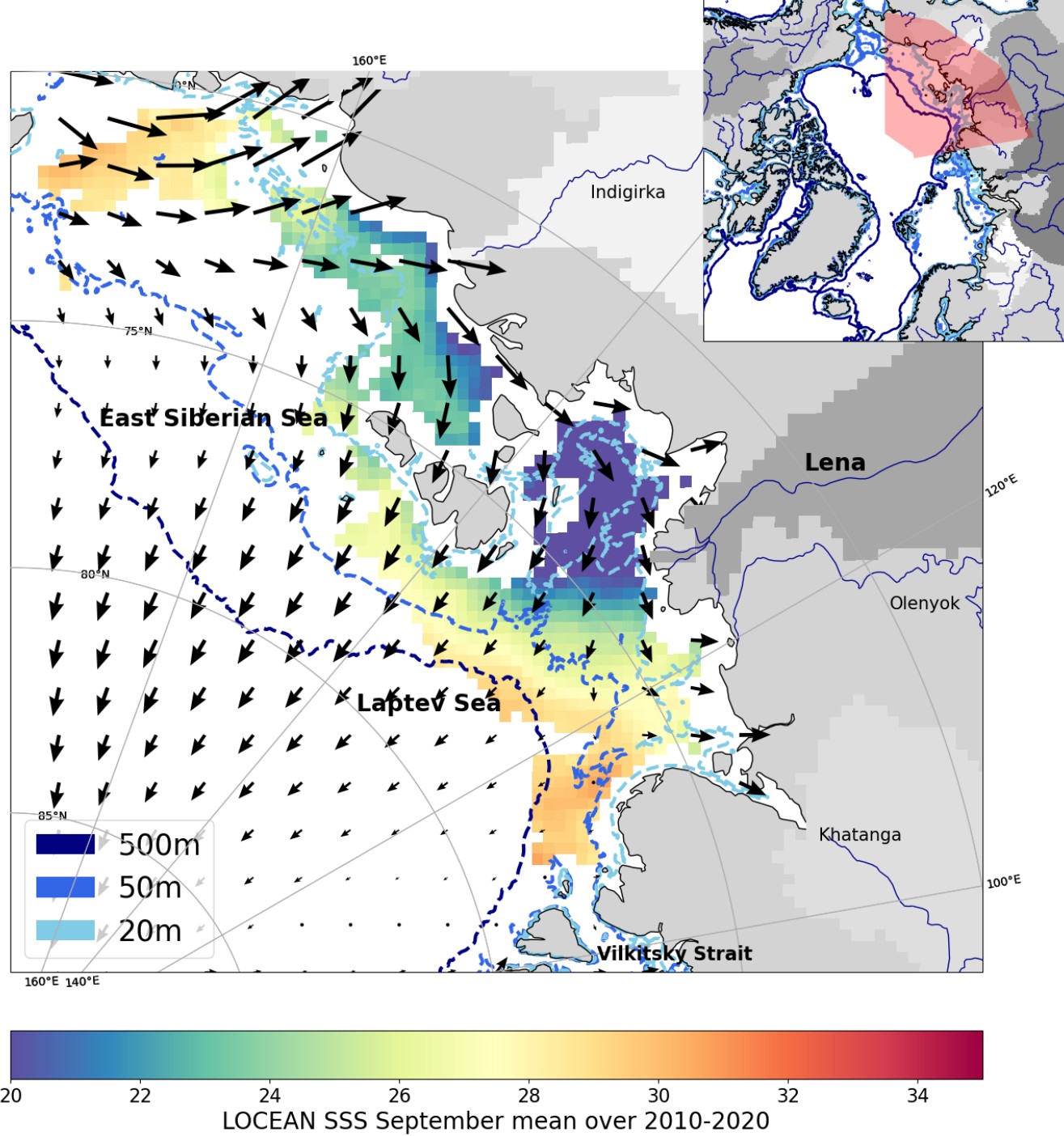

Figure 1: 2010-2020 LOCEAN SMOS satellite mean September SSS with GEBCO bathymetry contours for 20m, 50m and 500m overlaid in blue with mean 2010-2020 ERA5 June-August wind vectors overlaid over the ocean. The inset in the top right corner depicts Arctic wide GEBCO bathymetry and the location of this region within the wider Arctic in red.

The Laptev Sea primarily receives runoff from the Lena River, the largest river in the Arctic, which outflows as a particularly shallow plume due to the confined depth (~2-3m) of the Lena Delta (Are and Reimnitz, 2000). Lena River fresh water dominates the spatial pattern of Laptev sea surface salinity (SSS) and is the main control on stratification in this region (Janout et al., 2020). Lena runoff is very seasonal with very low flow throughout the winter, when the Lena River is partially frozen, and a strong peak between May and June following the melt of snow and land ice (Shiklomanov et al., 2021; Wang et al., 2021). Other rivers in this region, including the Khatanga, Olenyok and Indigirka, also contribute fresh water to the Laptev but all combined provide a five times smaller contribution than the Lena (Pasternak et al., 2022). Kara Sea fresh water can also contribute riverine fresh water to some of the western and northern Laptev shelf via the Vilkitsky Straight but contributions vary considerably interannually (Janout et al., 2020, 2015; Osadchiev et al., 2023). Sea ice melt also provides fresh water to the Laptev Sea but has a negligible impact in summer/autumn as the freshwater contribution from sea ice melt is several orders of magnitudes smaller than the contribution from the Lena River (Dubinina et al., 2017).

Laptev Sea surface fresh water is typically characterized by eastward (cyclonic) circulation and weak tidal influence (Fofonova et al., 2014; Timokhov, 1994). This fresh surface layer exhibits considerable interannual variability, varying in meridional extent by over 500km, and has been widely studied using in-situ data and model output (Anderson et al., 2004; Dmitrenko et al., 2005, 2008; Fofonova et al., 2014; Janout et al., 2020; Osadchiev et al., 2021). The shallow Laptev shelf (depth ~20-25m) is mostly controlled by wind forcing and bottom friction, and the strong stratification on this shallow shelf prevents a full Ekman spiral from developing and aligns the surface current ~45 degrees to the right of the wind (Dmitrenko et al., 2005; Kubryakov et al., 2016; Osadchiev et al., 2021; Zhuk and Kubryakov, 2021). Summer precipitation and sea ice melt contribute significantly less freshwater than rivers and are only suggested to provide a minimal direct contribution to altering summer SSS (Dubinina et al., 2017). River discharge variability has also been suggested as a driver of fluctuations in freshwater content and plume structure (Horner-Devine et al., 2015; Umbert et al., 2021). However, whilst there is general agreement that wind forcing is a driver of variability on the shelf, there is some debate as to the role of river discharge in controlling plume variability (Dmitrenko et al., 2005; Osadchiev et al., 2021).

Whilst Lena River water typically remains in the Laptev Sea for 2-3 years, its longer-term fate exhibits considerable variability as it can be transported out of the Laptev Sea either northward into the Transpolar Drift or eastward towards the Beaufort Gyre (Bauch et al., 2013; Johnson and Polyakov, 2001; Paffrath et al., 2021). Large-scale atmospheric circulation / the Arctic Oscillation Index (AOI) and the initial transport of the fresh layer have been suggested as the main controls on its eventual transit (Johnson and Polyakov, 2001; Morison et al., 2012).

In the Arctic, in-situ measurements of salinity have long been particularly sparse and infrequent due to the persistent sea ice cover that restricts access throughout most of the year. Satellite SSS has the potential to be an invaluable tool as salinity is the

dominant driver of density at high latitudes and plays a key role in controlling transport around the Arctic. However, sea ice and the low sensitivity of L-band signal in cold water has historically made satellite SSS retrievals at high latitudes a challenge. Recent progress in satellite product development has considerably lowered bias by over 0.15 pss compared to in-situ data in the Arctic, increasing confidence in acquisitions and making satellite SSS data a valuable resource for Arctic studies (Fournier et al., 2019; Supply et al., 2020). In addition, retreating Arctic sea ice cover and rapid atmospheric warming increases the spatial cover of satellite based SSS measurements. Whilst SSS retrievals at high latitudes still have larger uncertainties relative to the rest of the globe, previous works have shown that accuracy is sufficient to capture regions with sharp SSS gradients and demonstrated its potential for looking at Eurasian river plumes (Kubryakov et al., 2016; Olmedo et al., 2018; Supply et al., 2020; Tang et al., 2018; Zhuk and Kubryakov, 2021).

In this manuscript, we first establish that the reanalysis and satellite products used in this study (described in section 2.1) capture the interannual variability in Laptev SSS observed in in-situ data in section 3. The dominant drivers of this variability are then investigated using GLORYS12V1: including the contribution of Lena River runoff and of local and Arctic-wide atmospheric forcing in driving these patterns of variability. The findings of this analysis are then tested using satellite SSS data. A similar analysis is also conducted with sea surface temperature (SST) and sea ice concentration (SIC) data to understand common and differing drivers of variability and how the components of this system interact.

## 2 Data and Methods

### 2.1 Data Products

#### 2.1.1 In-situ data

CTD profiles from cruises in 2016 and 2019 are used for comparison with reanalysis data to study vertical salinity stratification in this region and to complement surface salinity data (supplementary materials (Osadchiev et al., 2021). Additional in-situ data from CTD probes, floats, ice-tethered profilers, oceanographic cruises and other platforms in the Laptev Sea are used for validation of satellite and reanalysis products from a number of sources (UDASH, NABOS cruises, and cruises on Akademik Mstislav Keldysh). See Appendix for details on in-situ data used to validate satellite SSS products (see Appendix A Table A1, Figure A1).

Lena River runoff data from the Arctic Great Rivers Observatory (GRO) dataset is used to identify the main drivers of Laptev Sea interannual variability (Shiklomanov et al., 2021). Cumulative runoff until a certain Julian day of each year is calculated for spring (Julian day 150), summer (Julian day 250) and the full year (Julian day 365). The spring peak in runoff has been shown to be shifting earlier with the rapidly warming Arctic (Yang et al., 2002; Melnikov et al., 2019) so a notable trend is

present in the spring cumulative runoff timeseries with the shift to earlier permafrost thaw / river ice melt. To avoid spurious

correlation and to be able to differentiate drivers of interannual variability from decadal/longer term trends, the trend in cumulative runoff timeseries (over the GLORYS time period of 1993-2022) is identified and removed. The de-trended spring runoff timeseries is used throughout this study.

### 2.1.2 Reanalyses

The 1/12 degree CMEMS GLORYS12V1 reanalysis (hereafter referred to as GLORYS12V1) (Lellouche et al., 2021) is used as a comparison dataset alongside the satellite products over the common observational periods (since 2011/2015) in this region (Table 1). This reanalysis is chosen for its high spatial resolution, its good representation of Arctic SIC and its previous application to salinity variability in the Subpolar North Atlantic and Arctic (Biló et al., 2022; Hall et al., 2021; Lellouche et al., 2021; Liu et al., 2022). For consistency with the satellite SSS products, the GLORYS12V1 reanalysis is re-gridded onto a

0.25° grid for comparison with in-situ data.

To demonstrate the benefit of using satellite SSS in this region, four ¼ degree reanalysis products (Table 1) are also validated against in-situ data (Appendix A Table A2, Table A3). These include: GLORYS2V4 from Mercator Ocean, ORAS5 from ECMWF, GloSea5 from Met Office, and C-GLORS05 from CMCC (Masina et al., 2017).


Table 1: Reanalysis products used in this study and their start and end dates, the number of vertical levels they have and native and used temporal and spatial grid resolutions

| Reanalyses | Start date used | End date used | Native temporal resolution | Temporal resolution used | Vertical levels | Native grid spatial resolution | Grid spatial resolution used |
|---|---|---|---|---|---|---|---|
| **GLORYS12V1** | 1993-01 | 2020-12 | Monthly | Monthly | 50 | 0.083° | 0.083°, 0.25° used only for validation |
| **GLORYS2V4** | 1993-01 | 2019-12 | Monthly | Monthly | 75 | 0.25° | 0.25° |
| **ORAS5** | 1993-01 | 2019-12 | Monthly | Monthly | 75 | 0.25° | 0.25° |
| **GloSea5** | 1993-01 | 2019-12 | Monthly | Monthly | 75 | 0.25° | 0.25° |
| **C-GLORS05** | 1993-01 | 2019-12 | Monthly | Monthly | 75 | 0.25° | 0.25° |

ECMWF's 5th generation reanalysis of global weather and climate (ERA5) monthly eastward and northward turbulent surface stress is used in assessing the main drivers of Laptev Sea interannual variability (Hersbach et al., 2020). The monthly mean NCEI Climate Prediction Center (CPC) Arctic Oscillation Index (AOI) (https://www.ncei.noaa.gov/access/monitoring/ao/) is also used to relate local eastward wind stress patterns to larger scale atmospheric circulation.

**2.1.3 Satellite data**

To validate and identify strengths and weaknesses of satellite-based SSS measurements over the Laptev Sea, this study uses two SMOS and two SMAP monthly products which are described below (Table 2). Higher temporal resolution satellite products were considered for analysis but comparison with in-situ data suggested they do not notably improve correlations with in-situ data. Therefore, these results do not justify their use over monthly products.


**Table 2: Satellite sea surface salinity products used in this study and their start and end dates, and native and used temporal and grid resolutions**

| SSS Products | Start date used | End date used | Native temporal resolution | Temporal resolution used | Native grid resolution | Grid resolution used |
|---|---|---|---|---|---|---|
| L3 LOCEAN SMOS Artic v1.1 | 2010-06 | 2019-12 | Monthly | Monthly | 25km EASE | 0.25° |
| L3 BEC SMOS ARCTIC+ v3.1 | 2011-01 | 2019-12 | 3day | Monthly | 25km EASE | 0.25° |
| L3 JPL SMAP v5 | 2015-04 | 2022-01 | Monthly | Monthly | 0.25° | 0.25° |
| L3 RSS SMAP v4 | 2015-04 | 2022-01 | Monthly | Monthly | 0.25° | 0.25° |

The two SMAP products are global products and are not specific for the Arctic: JPL (Jet Propulsion Laboratory) v5 and RSS
v4 (Remote Sensing Systems). Given the SMAP satellite's later launch, the SMAP products are compared over 2015-04 to 2022-01. The SMAP JPL product provides a large coverage including close to the sea ice edge. To be comparable with other products, data are masked to only include SSS where the SSS uncertainty provided in the product is lower than 1 pss. No masking is used for the three other products.

The two SMOS products are Arctic Ocean focused products: the L3 BEC (Barcelona Expert Centre) Arctic+ v3.1 and L3 LOCEAN (Laboratory of Ocean and Climatology) Arctic v1.1 products (Martínez et al., 2021; Supply et al., 2020). Monthly means are calculated from the 3-day BEC product to enable comparison with the other monthly satellite products. Their

common period of data availability is 2011-01 to 2019-12. The two SMOS products are regridded onto a regular 0.25° grid (consistent with the SMAP grid) for easier comparison with reanalysis and in-situ data.

SST measurements are taken from the gap-filled L4 CCI (Climate Change Initiative) SST CDR (Climate Data Record) v2.1 (Merchant et al., 2019). A monthly product of this data regridded at 0.1 ° resolution is used over the SSS satellite period (2010 to 2021).

## 2.2 Methods

We focus on September as the month of maximum open water area and hence the largest area of satellite and in-situ data for comparison with reanalysis products. Two Septembers are shown for comparison of how well interannual variability is captured in each satellite product: 2016, a year of predominant eastward wind and 2019, a year of predominant westward wind (Figure 2, Figure 3). This study does not consider variability in SSS below 20 pss due to the sparsity of in-situ observations with SSS values below this threshold. As is shown in section 3.1, JPL SMAP and LOCEAN SMOS are found to agree particularly well so are used for further analysis.

To investigate the contribution of key drivers to Laptev Sea interannual variability, a lagged correlation analysis is conducted between GRO runoff, ERA5 eastward turbulent surface stress and GLORYS12V1 SSS, SST and SIC over the full GLORYS12V1 timeseries (1993-2020). Pearson correlation coefficients are calculated between cumulative GRO runoff until spring (Julian day 150), summer (Julian day 250) and over the full year (Julian day 365) and GLORYS12V1 September SSS for each grid cell (Figure 4). The same correlations are calculated with GLORYS12V1 September SST and SIC at each grid cell (Figure 5, Figure 6).

Pearson correlation coefficients were also calculated between ERA5 eastward turbulent surface stress in April, May, June, July, August and September and GLORYS12V1 September SSS (over 1993-2020) in each grid cell to identify the months that appear to most strongly drive variability in September SSS (Figure 4). The same correlations are calculated over the same time period (1993-2020) with GLORYS12V1 September SST and SIC at each grid cell (Figure 5, Figure 6).

To identify years of anomalous eastward/westward wind over the shorter satellite timeseries, the mean ERA5 eastward and northward turbulent surface stress are calculated for June to August over the Laptev Sea shelf: 120-160 °E, 70-80 °N. The period of June to August is chosen because of the particularly strong correlations found in the lagged correlation analysis between eastward turbulent surface stress in June, July and August and GLORYS12V1 September SSS (Figure 4). A three-

month mean is chosen to reduce the high temporal variability in wind stress ($\pm$ 0.05 N m$^{-2}$) and only keep the lower frequency signal the ocean reacts to.

The mean AOI for June to August is calculated for comparison with Laptev Sea eastward turbulent surface stress to relate local
wind stress to large-scale atmospheric circulation (Figure 7). The correlation between local eastward turbulent surface stress and the AOI is calculated over the satellite timeseries (2010-2022) and over a longer timeseries (1993-2022). Correlations are also calculated between spring runoff and eastward turbulent surface stress and AOI over the satellite timeseries (2010-2022) and over a longer timeseries (1993-2022). Correlations were also calculated over a longer timescale (1993-2022) to ensure robustness and consistency of correlations found.


To be able to calculate "eastward" and "westward" SSS and SST composites, the 3 years of maximum and minimum eastward turbulent surface stress are identified for each of the two satellite periods (SMOS: 2011-2020 and SMAP : 2015-2022). The three years of maximum eastward turbulent surface stress are identified to be 2012, 2016 and 2017 over the SMOS timeseries, and identified to be 2016, 2017 and 2021 over the SMAP timeseries (Figure 7). Conversely, the three years of westward
(minimum eastward) turbulent surface stress are identified to be 2011, 2013 and 2019 over the SMOS timeseries, and 2015, 2019 and 2020 over the SMAP timeseries.

The "eastward" SSS composite is then calculated as the mean of the three most eastward years for GLORYS12V1 SSS and LOCEAN SMOS (2012, 2016, 2017), and for JPL SMAP (2016, 2017, 2021). The "westward" SSS composite is calculated
as the mean of the three most westward years for GLORYS12V1 SSS and LOCEAN SMOS (2011, 2013, 2019) and for JPL SMAP (2015, 2019, 2020). The same years are used to calculate "eastward" and "westward" SST composites using GLORYS12V1 SST and L4 v2.1 CCI SST as well as for GLORYS12V1 SIC.

# 3 Results:

## 3.1 Comparison of SSS products

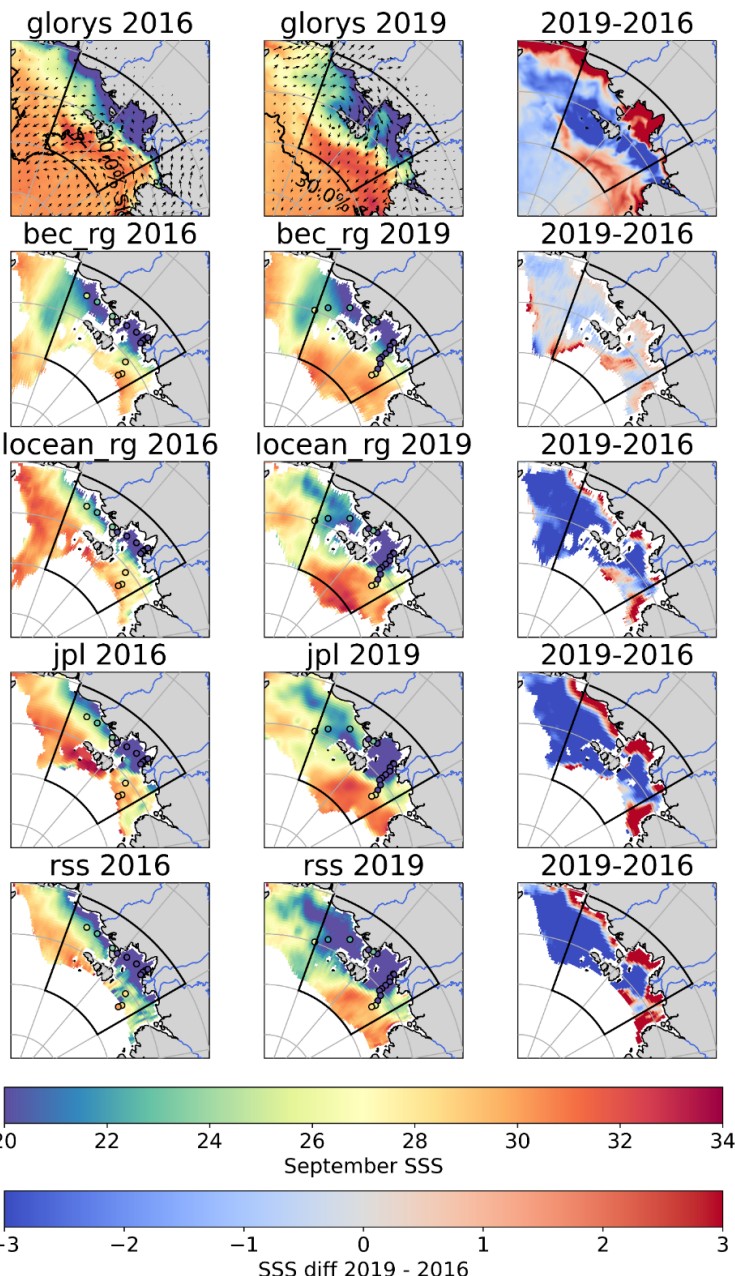

**Figure 2: Laptev Sea sea surface salinity field in September (9) 2016 (left) and 2019 (middle) and the difference between 2016 and 2019 (right) for the CMEMS GLORYS12V1 reanalysis (top) and for each of the 4 satellite products (RSS SMAP, JPL SMAP, LOCEAN SMOS, BEC SMOS) (top to bottom). ERA5 mean wind speed for June-August are overlaid on the GLORYS12V1 SSS field with a box over the region of interest (70-80°N, 120-160°E). The GLORYS12V1 30% sea ice concentration contour is also overlaid as a black line over the GLORYS12V1 SSS field. In-situ data for late September 2016 and early October 2019 are overlaid on satellite products using the same colour scale.**

There is close agreement between the September SSS pattern in GLORYS12V1 and all satellite September SSS products in both years compared (Figure 2). The SSS off the continental shelf (> 100 m) or above 75 °N is typically > 28 pss in both years analysed and in both products. SSS generally decreases with proximity to shore, and is lowest near the outflow of the Lena River, around 130 °E, with salinity values as low as 10 pss nearshore. This low salinity area (< 20 pss) extends considerably to the East of the Lena River outflow throughout the southern Laptev Sea and past the New Siberian Islands into the East Siberian Sea, extending to over 160 °E in both years.

The years 2016 and 2019 stand out as having notably different patterns of Laptev SSS, with differences in SSS of over 3 pss between years in all satellite products. GLORYS12V1 SSS and all satellite products except BEC capture the same SSS patterns as in-situ data from cruises in all years of overlap (2016 and 2019 shown in Fig. 1). In 2016, the freshest salinities are coastally confined and do not travel far off the continental shelf. In 2019, the freshest salinities travel considerably further offshore and extend over most of the Western Laptev and East Siberian Sea.

Despite the strong overall similarity between gridded products, notable differences are visible between in-situ data and both the satellite products and GLORYS12V1 SSS. In 2019, the fresh layer appears to extend further offshore in in-situ data than in GLORYS12V1 (Figure 2). LOCEAN, JPL and RSS appear to capture this extended plume better, but still do not capture the full extent visible in in-situ data. This difference is likely primarily due to the temporal mismatch between the September monthly mean GLORYS12V1 and satellite products and in-situ data collected in late September 2016 and early October 2019. Both GLORYS12V1 and satellite SSS do show the plume extending further offshore by the following month (not shown), supporting this suggestion. However, the better representation of plume extent in LOCEAN, JPL and RSS, as compared to GLORYS12V1, suggests the temporal mismatch is not the only driver of this difference.

Most of the satellite products (LOCEAN SMOS and both SMAP products) and GLORYS12V1 manage to capture a consistent pattern of interannual variability and agree well with in-situ data (Figure 2, Appendix A Figure A5, Figure A6). However, notably different patterns are observed in the BEC product, which also has a lower correlation with in-situ data (r = 0.79, Appendix A Table A2). All other satellite products analysed appear to capture the SSS pattern described above for 2016 and 2019 and correlate strongly with in-situ data (r > 0.9, Appendix A Table A2). This difference in SSS pattern agrees well with the two modes of SSS variability previously observed in in-situ data and described by other studies in this region (Dmitrenko et al., 2005; Osadchiev et al., 2021). Of the four products considered here, the LOCEAN SMOS Arctic and JPL SMAP products capture particularly consistent patterns of interannual variability and have strongest correlations with in-situ data (r=0.92 for LOCEAN, r=0.95 for JPL, Appendix A Table A2). This is notable given they originate from different satellites and are generated from different processing algorithms. These two products (LOCEAN SMOS and JPL SMAP) are further used in this study, for their strong similarity and good correlation values with in-situ data.

GLORYS12V1 and the two satellite products show similar areas of open water / of no retrievals (Figure 2). In 2019, the area of open water is particularly large in GLORYS12V1 and in all satellite products, with no regions of notable sea ice (where SIC > 30%) below 80 °N throughout the Laptev and East Siberian Seas. In 2016, there is more extensive sea ice and few satellite SSS retrievals in the Laptev Sea but a large area of open water in the East Siberian Sea, which extends considerably

offshore to over 80 °N.

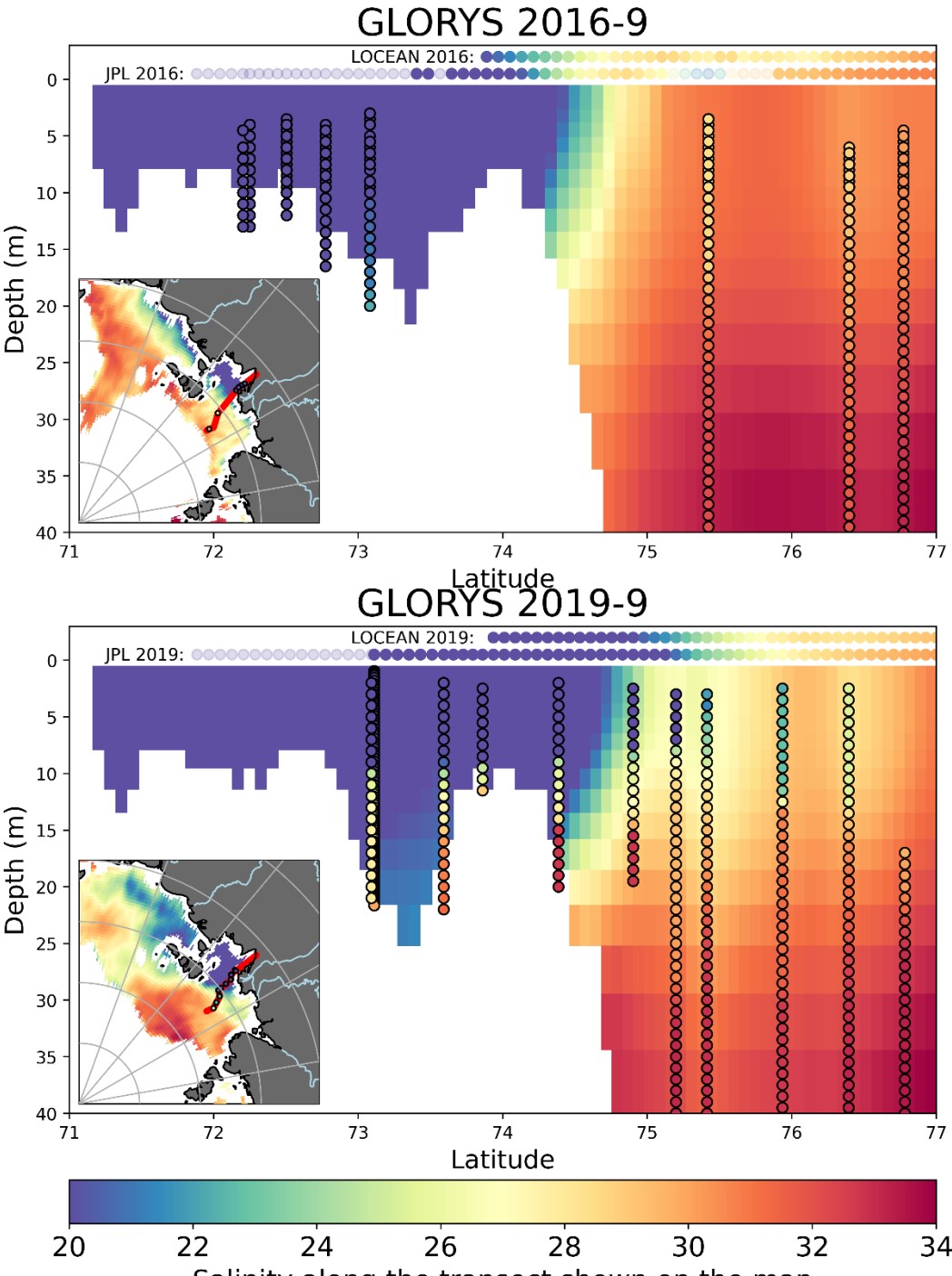

**Figure 3**: *GLORYS12V1 SSS vertical transect in 2016 (top) and 2019 (bottom) along red transect interpolated through in-situ data (shown in map of JPL SMAP SSS in bottom left for each year) with in-situ data overlaid with black rings and satellite data for that transect in JPL SMAP and LOCEAN SMOS SSS shown as a line of points. JPL SMAP data is made semi-transparent where the provided SMAP SSS uncertainty is < 1 pss.*

GLORYS12V1 features a well-mixed plume in the shallowest regions of the shelf in 2016 and 2019 (Figure 3), and in almost all other years considered. Hence, it agrees well with in-situ data in regions and years where the plume is well-mixed nearshore (e.g. 2016 shown above and 1994 and 2000 not shown) but fails to represent years with a stratified plume nearshore (e.g. 2019 shown above and 2008 and 2011 not shown). The variability of stratification dynamics, even just in the two years examined, suggests it is not appropriate to assume a constant mixed layer depth on the shallow shelf, as previously applied to estimate
fresh water content (Umbert et al., 2021). In all years examined, in-situ data shows the fresh layer (< 15 pss) is relatively shallow and only extends between 5 and 10m, shallower than Kara due to weaker tidal mixing (Osadchiev et al., 2021).

In 2019, differences in surface plume extent are visible between GLORYS12V1 and in-situ data. Some of these differences are due to spatio-temporal mismatch of September monthly 1/12 degree data with point in-situ data (in late September/early
October), as vertical stratification is very seasonally and regionally variable (and bathymetrically controlled) in this region (Janout et al., 2020). However, both satellite products more closely resemble the extended plume visible in in-situ data than GLORYS12V1.

In addition, previous studies show considerable interannual variability in the lowest values of SSS at the outflow of the Lena
River. Whilst in certain years, there are only very small regions of SSS below 20 pss (2014), in other years, notable regions of SSS as low as 6 pss have been observed (in 2013) (Janout et al., 2020). Within GLORYS12V1, the shallow surface layer is consistently more saline (between 15-20 pss) than in-situ data and salinities below 20 pss are typically very confined to the shelf. Although there are few satellite SSS retrievals near the coast (due to land contamination), nearshore SSS are notably lower and quite variable (10-20 pss) in LOCEAN SMOS and JPL SMAP and more consistent with in-situ data. Overall, within
shallow shelf regions (< 20 m), the more saline surface waters, fresher subsurface waters and less extensive surface plumes suggest GLORYS12V1 is too well-mixed compared to in-situ data. This is reinforced by the weak tidal influence in this region and as there is rarely sufficient wind-driven mixing to break up such strong stratification (Fofonova et al., 2014; Hölemann et al., 2011; Janout and Lenn, 2014; Shakhova et al., 2014).

Salinity stratification on the shelf is much stronger than that of temperature and is by far the dominant control on density in this region (Osadchiev et al., 2021). SST, and in turn stratification in temperature also vary considerably over the course of September, so a higher temporal resolution analysis would be needed for investigating temperature stratification dynamics. This is visible from the difference between in-situ data (from late September/early October) and September mean satellite/reanalysis SST data (Appendix B Figure A2). Therefore, this study focuses on salinity stratification in this region,
which is more consistent over the course of September, and more appropriately represented by the monthly data used for analysis in this study.

**3.2 Impact of runoff and wind stress on SSS, SST and SIC in GLORYS12V1**

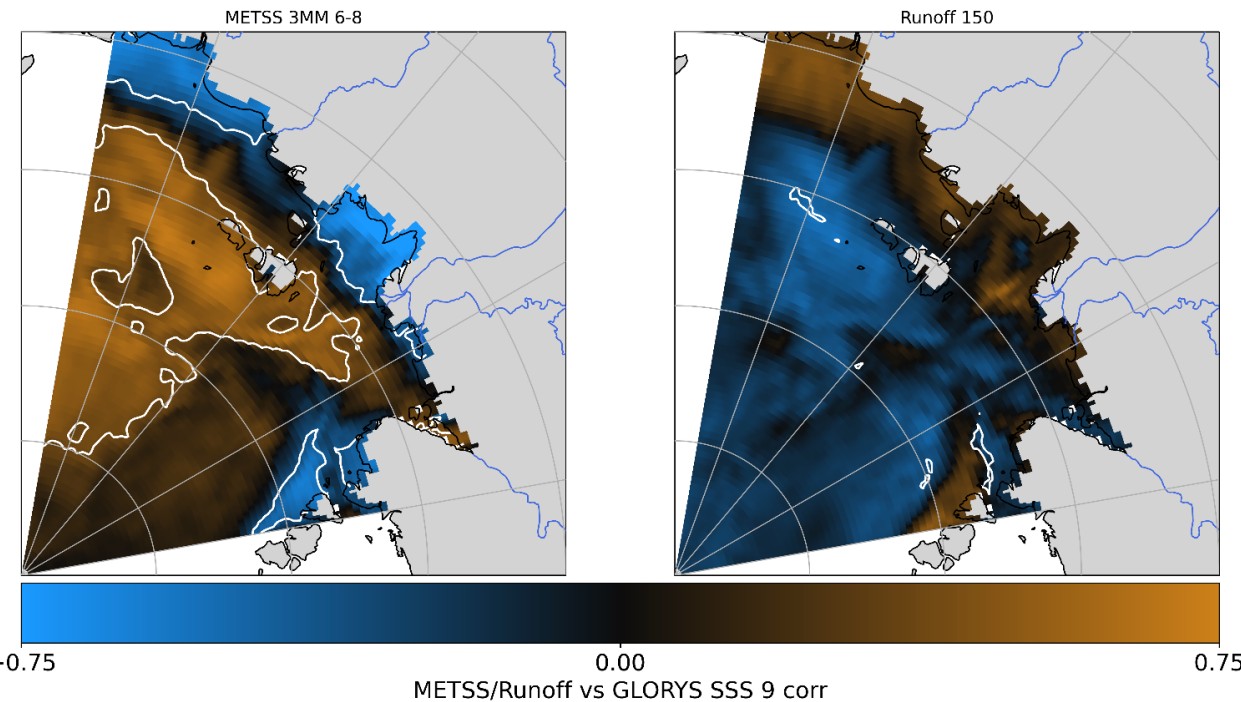

**Figure 4: Correlation between GLORYS12V1 September SSS and the three-month mean ERA5 eastward turbulent surface stress (METSS) over June to August (6-8) (left) over 1993-2022. Correlation between GLORYS12V1 September SSS and cumulative Lena River runoff in spring (Julian day 150) (right) over 1993-2022. Regions where correlations are statistically significant (p≤0.05) are denoted by the white contour and brighter colours.**

There is a significant spatial pattern in correlation between the three-month (June to August) mean eastward turbulent surface stress and GLORYS12V1 September SSS field for the 1993-2022 time period (Figure 4). This pattern consists of a strong negative correlation nearshore (< -0.75) and a strong positive correlation offshore (> 0.75), particularly in the East Siberian Sea. The negative correlation suggests strong eastward wind stress is consistent with fresher SSS nearshore. The strong positive correlations offshore are present, albeit in different regions, throughout June, July and August, as well as in the three-month mean (Appendix B Figure A3). However, the negative correlation nearshore is only present in July and August. A small region of negative correlation (< -0.75) is also present just East of the Vilkitsky Strait, and is visible in all three months. These strong correlations are statistically significant at $p < 0.05$ (highlighted by the white contour).

A weak, non-significant spatial pattern in correlation is found between cumulative spring runoff and GLORYS12V1 SSS. This pattern suggests a positive correlation nearshore, particularly in the East Siberian Sea, and a negative correlation offshore. The weak positive correlation nearshore suggests increases in runoff are consistent with increases in SSS. This pattern is the opposite of what would be expected and what has previously been suggested: that an increase in runoff would drive nearshore

freshening. However, there is almost nowhere that this correlation is statistically significant. Whilst there are some regions that yield significant p values, these regions are all very small and do not appear to depict a relationship with a physical basis. The spatial patterns of correlation between GLORYS12V1 September SSS and both cumulative summer and total annual runoff show similar correlations but are even weaker and are not statistically significant (Appendix B Figure A3).

340

Similar correlation analyses conducted between runoff and eastward surface stress with SSS in the other reanalysis products yielded similar spatial correlation patterns to those visible here in GLORYS12V1.

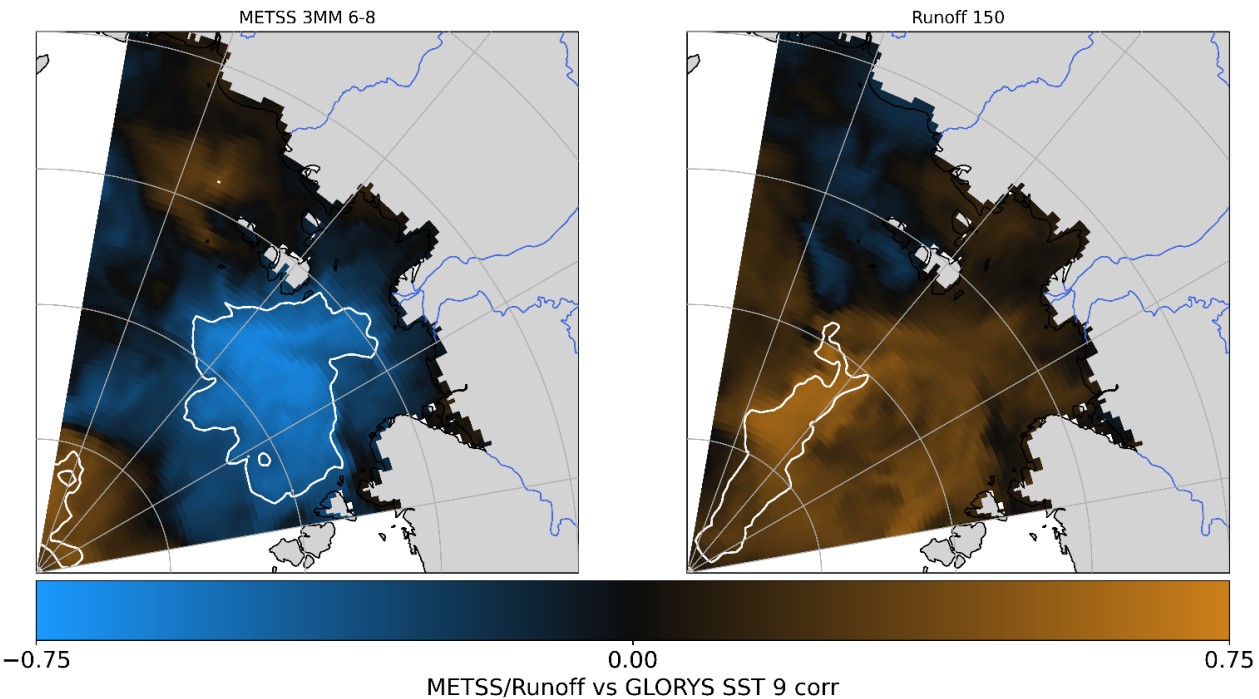

**Figure 5: Correlation between GLORYS12V1 September SST and the three-month mean ERA5 eastward turbulent surface stress (METSS) over June to August (6-8) (left) over 1993-2022. Correlation between GLORYS12V1 September SST and cumulative Lena River runoff in spring (Julian day 150) (right) over 1993-2022. Regions where correlations are statistically significant (p≤0.05) are denoted by the white contour and brighter colours.**

Figure 5 depicts the strong spatial pattern of correlation between the three-month (June to August) mean eastward turbulent surface stress and GLORYS12V1 SST. This spatial pattern consists of strong negative correlations (< -0.75) near the edge of the continental shelf. This negative correlation suggests eastward wind stress is consistent with cooler SSTs near the edge of the continental shelf (and that westward wind stress is consistent with warmer SSTs in this region). The region of negative correlation differs in region between June and August (Appendix B Figure A4). It is closest to shore in June and appears to

move offshore over July and August. Whilst this negative correlation is mainly confined to the Laptev Sea in June and August, it extends into the East Siberian Sea in July. No significant correlation is present nearshore in any month.

A weak positive correlation is present between cumulative spring runoff and GLORYS12V1 SST throughout the Laptev Sea. This positively correlation is not statistically significant anywhere except in the central Arctic (> 80 °N). This positive correlation suggests increased spring runoff is consistent with warmer SSTs throughout the Laptev Sea. The spatial patterns of correlation between GLORYS12V1 September SST and both cumulative summer and total annual runoff show similar correlations but are even weaker and are not statistically significant (Appendix B Figure A4).

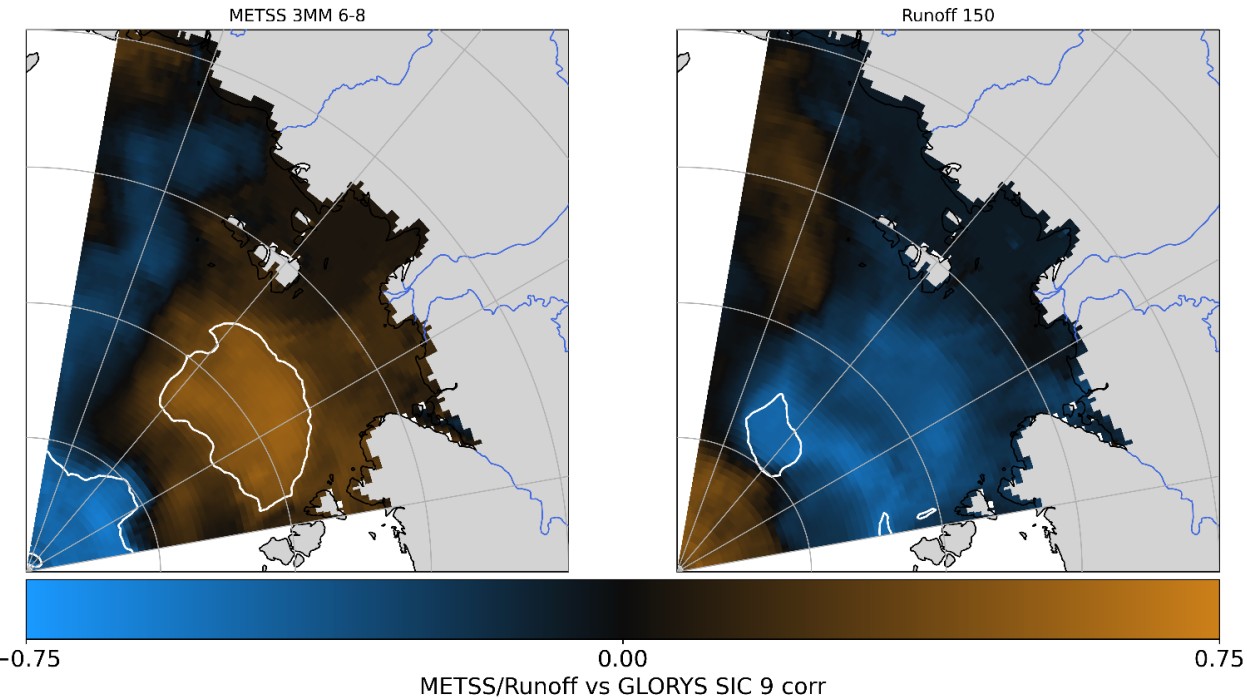

**Figure 6: Correlation between GLORYS12V1 September SIC and the three-month mean ERA5 eastward turbulent surface stress (METSS) mean over June to August (6-8) (left). Correlation between GLORYS12V1 September SIC and spring cumulative Lena River runoff (to Julian day 150) (right) over 1993-2022. Regions where correlations are statistically significant (p≤0.05) are denoted by the white contour and brighter colours.**

Figure 6 depicts the strong spatial pattern of correlation between the three-month (June to August) mean eastward turbulent surface stress and GLORYS12V1 SIC. This pattern suggests a large region of strong positive correlation is present just off the continental shelf in the Laptev Sea and a region of strong negative correlation is present in the central Arctic (> 85 °N). In turn, this implies that eastward wind stress is consistent with increased SIC in the northern Laptev Sea and lower SIC in the central Arctic (and that westward wind stress is consistent with decreased SIC in the northern Laptev Sea and increased SIC in the central Arctic).

A large region of weak negative correlation is present between cumulative spring runoff and GLORYS12V1, suggesting
       increased spring runoff is consistent with lower SIC. This negative correlation is present throughout almost all the Laptev Sea
       but is only significant ($p < 0.05$) near 84 °N.

## 3.3 Drivers of interannual variability in September SSS

The mean atmospheric circulation pattern is represented in Figure 7, calculated as the mean surface stress over the box defined
       in Figure 2. Values are notably different in 2016 and 2019 (Figure 2). In 2016, there is predominantly cyclonic circulation,
       with strong Eastward winds dominant over the Laptev Sea shelf, and Northward winds present over the region of the Laptev
       Sea just off the continental shelf. In 2019, there is predominantly anticyclonic circulation with North-westward winds dominant
       over the Laptev Sea shelf. The anticyclonic circulation visible in 2019 more closely resembles the mean circulation pattern
visible over 2011-2020 (Figure 1).

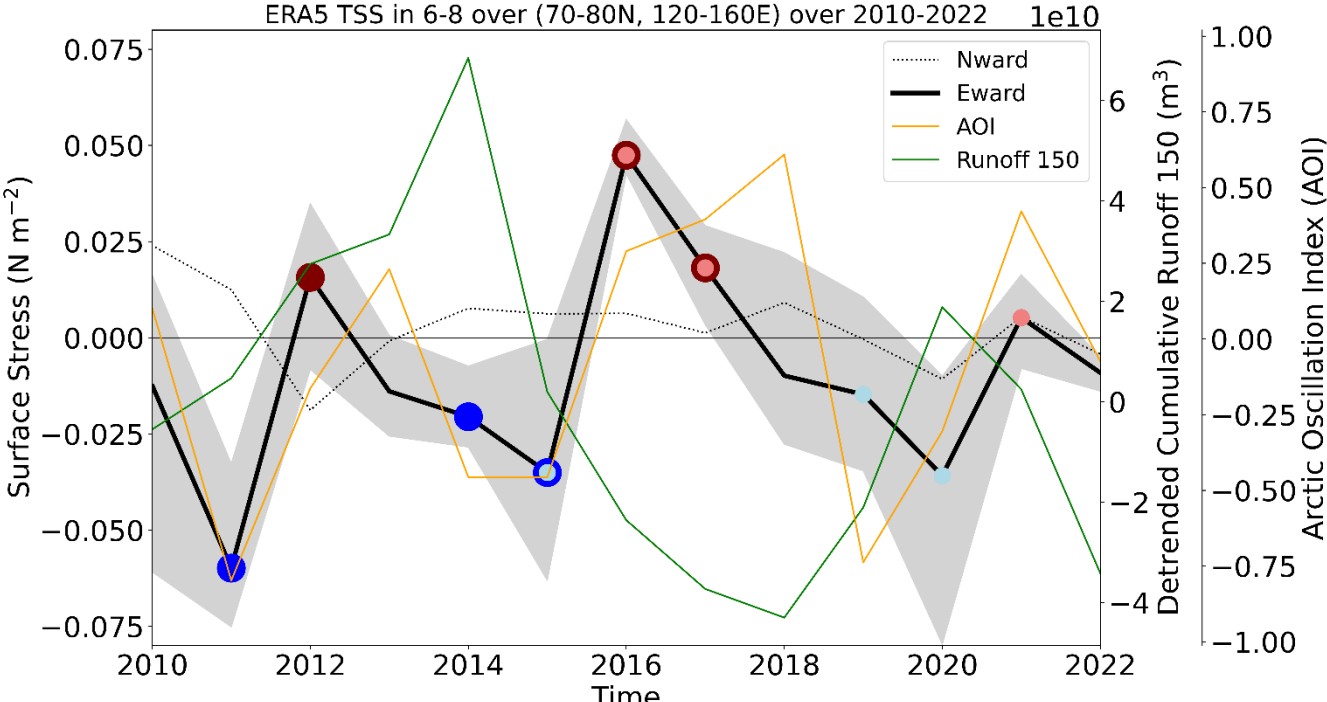

**Figure 7: Three-month (June to August) mean ERA5 eastward (black solid) and northward (black dashed) turbulent surface stress
over 70-80 North and 120-160 East. Overlaid are dots indicating the most eastward (red dots) and westward (blue dots) years chosen
for analysis for both the longer SMOS/GLORYS12V1 timeseries (2011-2020) (darker red and blue dots) and the shorter SMAP
timeseries (2015-2022) (lighter red and blue dots). The range of the maximum and minimum eastward turbulent surface stress
between June and August is shaded in grey. Spring cumulative Lena River runoff (until the 150[th] Julian day) (green) and mean June
to August arctic oscillation index (AOI) (orange) are overlaid.**

The magnitude of variability in mean eastward turbulent surface stress ($\pm 0.05$ N m$^{-2}$) across the entire timeseries is notably larger than that of northward turbulent surface stress, which remained within $\pm 0.02$ N m$^{-2}$. The years of highest eastward turbulent wind stress are 2012, 2016, 2017 over the SMOS timeseries and 2016, 2017 and 2021 over the SMAP timeseries. The years of strongest westward turbulent wind stress are 2011, 2013 and 2019 over the SMOS timeseries and 2015, 2019 and 2020 over the SMAP timeseries. In years where the mean eastward turbulent surface stress is negative (denoting predominant westward turbulent surface stress), there is considerably more within-year variability (typically > 0.05 N m$^{-2}$ in eastward turbulent surface stress in the months spanning June to August (denoted by the grey overlay in Figure 7).

There is good agreement between the three-month mean of eastward turbulent surface stress and AOI over either 2010-2022 (r = 0.65, p = 0.02) or 1993-2022 (r = 0.49, p = 0.01). The AOI is highest in 2016, 2017 and 2018 over the SMOS timeseries and 2017, 2018 and 2021 over the SMAP timeseries. The AOI is lowest in 2011, 2015 and 2019 over the SMOS timeseries and 2015, 2019 and 2020 over the SMAP timeseries.

Spring cumulative runoff does not significantly co-vary with turbulent surface stress over 2010-2022 (r=-0.30, p=0.31) or over 1993-2022 (r = -0.29, p = 0.11). Spring cumulative runoff also does not significantly co-vary with the AOI over 2010-2022 (r = -0.39, p = 0.19) or over 1993-2022 (r = -0.20, p = 0.27). Spring runoff is highest in 2012, 2013 and 2014 over the SMOS timeseries and 2015, 2020 and 2021 over the SMAP timseries. Spring runoff is lowest in 2016, 2017 and 2018 over both the SMOS and SMAP timeseries. Interannual variability in runoff, turbulent surface stress and in the AOI over the short satellite period visible in Figure 7 are consistent with interannual variability over longer time periods (not shown).

415

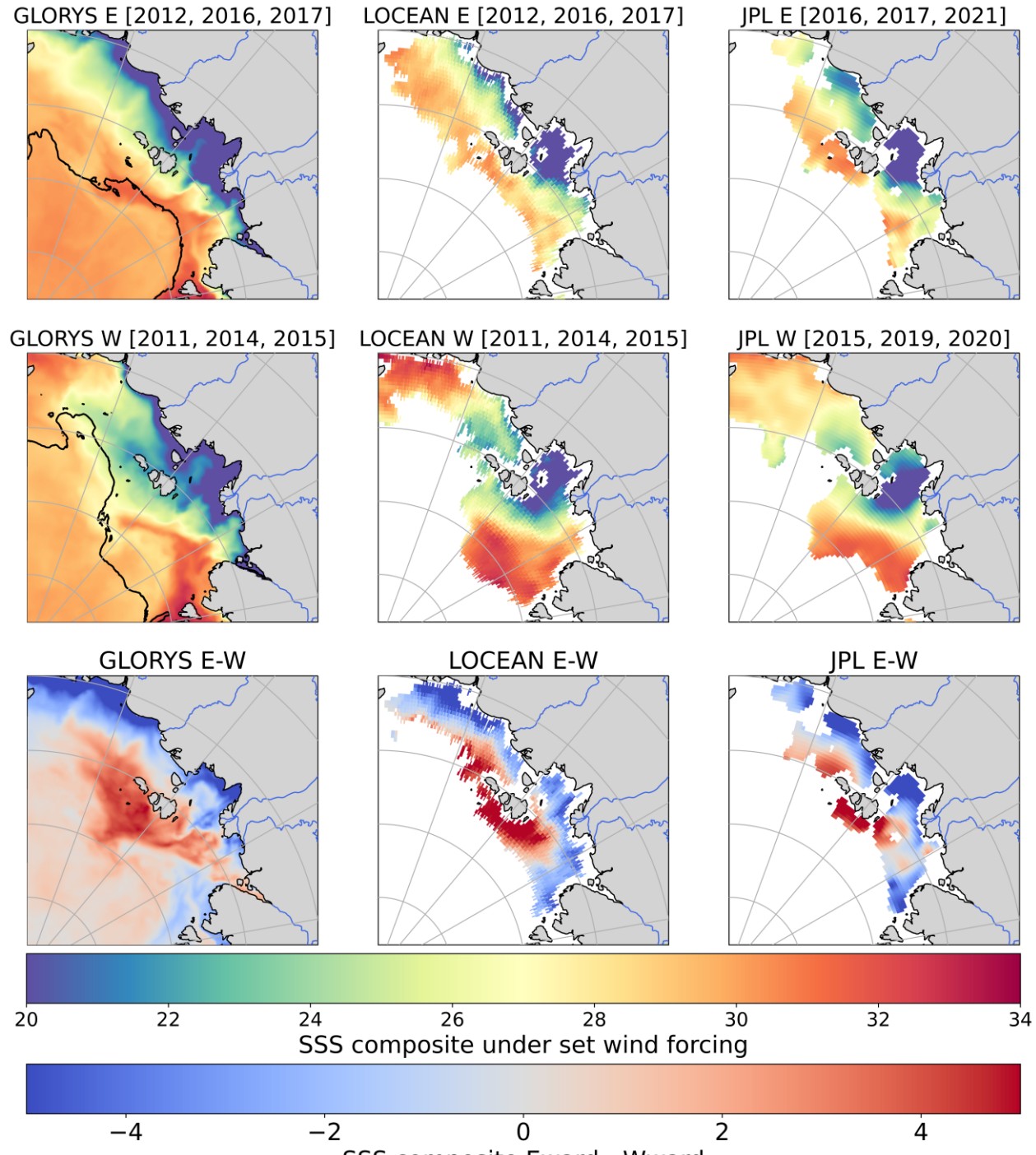

**Figure 8:** *Eastward (E, top row) and westward (W, middle row) composites calculated for (left to right) GLORYS12V1 SSS, LOCEAN SMOS and JPL SMAP, from the identified three most eastward and westward years (over 2011-2020 for GLORYS12V1 and LOCEAN SMOS and over 2015-2022 for JPL SMAP). The difference composite (eastward – westward) for each product is shown on the bottom row. The GLORYS12V1 mean 30% sea ice concentration contour is overlaid on the respective composite plots.*

The eastward/westward composites of all three SSS products agree strongly, regardless of the differing years chosen for analysis (Figure 8). The composite analysis highlights the differing pattern of SSS under positive (eastward) and negative (westward) zonal wind. The eastward composite closely resembles the 2016 SSS pattern visible in Figure 2, and the westward composite closely resembles the 2019 SSS pattern. This strong resemblance between particularly anomalous individual years and the zonal wind composite plots supports that the zonal wind is the dominant driver of variability in this region. Years with strong westward wind have considerable offshore transport, and northward spreading of the plume, denoted by the presence of anomalous fresh water in the Northern Laptev Sea and relatively higher salinity water in the East Siberian Sea. Alternatively, years of eastward wind are associated with onshore and alongshore transport, and a coastally confined plume, denoted by more saline waters in the Northern Laptev Sea and fresher waters in the Southern Laptev and East Siberian Seas.

The composite difference plots provide a clearer visualisation of the North/South (offshore/nearshore) dipole in freshwater transport visible under eastward/westward wind forcing. The strong agreement between all three products strengthens the weighting of this finding, particularly as the difference plots appear to agree even more closely than the individual eastward/westward composites. This agreement suggests that although the three products have different mean SSS states, they capture very similar patterns of variability.

There is a notable difference in SIC in years of westward and eastward wind forcing in both GLORYS12V1 and the satellite data (indicated by the absence of SSS data). Under westward wind forcing, the Laptev SIC is smaller in the Laptev Sea and the 30% SIC contour is nearer shore in the East Siberian Sea. The opposite is true under eastward wind forcing, with a larger SIC in the Laptev Sea and the 30% SIC contour further offshore in the East Siberian Sea.

**3.4 Impact of variability in wind forcing on SST**

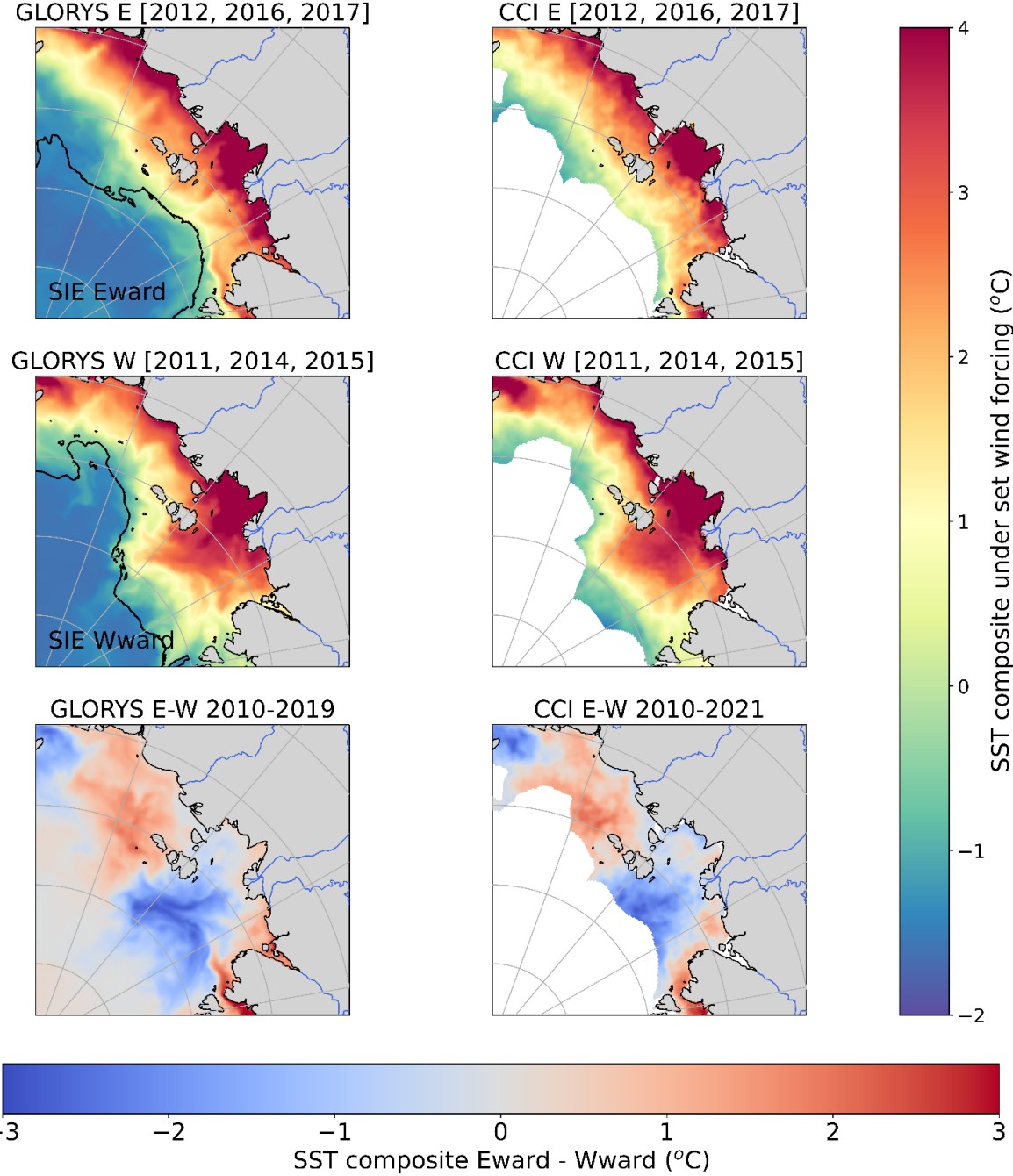

Figure 9: *Eastward (E, top row) and westward (W, middle row) composites calculated from the identified three most eastward and westward years over 2011-2020 for (left to right) GLORYS12V1 SST and L4 CCI SST (masked by 30% sea ice concentration). The difference composite (eastward – westward) for each product is shown on the bottom row. The mean 30% sea ice concentration contour for eastward and westward years is used to mask L4 CCI data and is overlaid in GLORYS12V1 in black on both eastward and westward composite plots.*

Similar to Figure 8, Figure 9 represents the eastward and westward composites of GLORYS12V1 and ESA CCI SST (Figure 9). Temperatures < 1 °C are typically present off the continental shelf in both composites (and all years analysed), with a rapid transition in temperature present at the 30% SIC margin (Figure 9). On the shelf, temperatures are typically warmer (> 1 °C) and riverine plume is typically > 2 °C, with large regions in excess of 4 °C.

The eastward/westward composites of both products agree very well and suggest notable differences in SST pattern under differing zonal wind forcing. Under eastward wind forcing, both GLORYS12V1 and CCI SST composites show that warm SST anomalies are confined to the southern Laptev Sea and travel alongshore towards the East Siberian Sea. This eastward wind state is coincident with a larger SIC in the Laptev Sea and a 30% SIC contour nearer shore in the East Siberian Sea. Under westward wind forcing, both SST composites show warm SST anomalies are mostly advected offshore to the Northern

Laptev Sea. The westward wind state is coincident with lower SIC in the Laptev Sea and a 30% SIC concentration contour further from shore in the East Siberian Sea. A dipole composite pattern is also visible in the SST difference composite, as is visible in the SSS difference composite. However, the difference composite between eastward and westward wind states presents in an East/West direction rather than a North/South direction.

**4 Discussion**

**4.1 Runoff as a driver of SSS, SST and SIC variability**

Spring, summer and annual Lena River runoff do not appear to play a role in controlling GLORYS12V1 September SSS, SST or SIC in the Laptev or East Siberian shelf seas. Cumulative spring runoff is most strongly correlated to variability in SSS, SST and SIC, suggesting the timing of the initial peak in runoff has more of an impact on Laptev Sea dynamics than the

cumulative runoff in summer or the total runoff over the year. However, the correlations with spring cumulative runoff are almost entirely not significant.

It might be expected that years with the largest magnitude of cumulative spring / summer / annual river discharge would have the largest fresh surface layer (< 20 pss) as previously suggested of cumulative annual discharge (Umbert et al., 2021).

However, the GLORYS12V1 correlation analysis suggests no significant correlation near the outflow of the Lena River between SSS and cumulative runoff at any time of year. If anything, the opposite pattern appears true nearshore in the East Siberian Sea: with increases in spring runoff driving higher salinities near the coast and low salinities offshore. The differing result here, compared to Umbert et al. (2021) appears to be linked to differences in BEC and GLORYS SSS, and the variability in fresh surface layer area. No alternative mechanism to date explain the opposing behaviour observed here. It is possible there

is some negative feedback whereby earlier spring runoff drives earlier sea ice retreat, and in turn expands the region of wind influence and spreads the plume further offshore. However, this counterintuitive correlation warrants further investigation.

The short nature of the satellite SSS timeseries prevents an in-depth correlation analysis with runoff as was done with GLORYS12V1 but visual comparison indicates no clear pattern between interannual variability in spring runoff and SSS over the SMOS or SMAP satellite periods. This comparison is also complicated by the interannually varying ice-free region, which determines the total area of SSS retrievals and in turn, any derived fresh surface layer areas. Over this period, spring runoff is lowest in 2016, 2017 and 2018. Whilst the fresh surface layer is extensive in 2018, it is very small and coastally confined in 2016 and 2017. These two years were excluded from analysis in Umbert et al. (2021) due to lack of SSS data, which partially explains the differing results here. Whilst the area of satellite SSS retrievals is relatively small in these years compared to other years analysed, the edge of the plume is clearly visible in the area of open water. This suggests that the small plume observed in these two years is not just due to the relatively small area of open water and that in turn, there is no reason to exclude these years from analysis. Conversely, spring runoff is highest in 2012, 2013 and 2014. Again, there is no conclusive SSS pattern as the fresh surface layer is relatively average in all three of these years. The inconsistent response in satellite data suggests cumulative spring runoff is not a major driver of interannual variability in SSS pattern, as is suggested from GLORYS12V1 and as has previously been suggested by other studies (Osadchiev et al., 2021).

## 4.2 Wind variability as driver of SSS variability

Previous studies using sparse in-situ data have suggested wind forcing appears to drive some variability in freshwater transport (Dmitrenko et al., 2005; Osadchiev et al., 2021). Satellite SSS data shown here provides a picture of SSS variability and confirms what has previously only been suggested from in-situ data: that zonal wind forcing is the dominant driver of Laptev SSS. Satellite SSS data also provides a clear, complete visualization of differences in freshwater transport throughout the sea ice free Laptev and East Siberian seas under different wind regimes, augmenting the scattered view available from in-situ data. Westward wind drives considerable offshore transport, and northward spreading of the plume toward the Northern Laptev Sea. Conversely, eastward wind is found to drive alongshore transport, resulting in a coastally confined river plume, denoted by more saline waters in the Northern Laptev Sea and fresher waters in the Southern East Siberian Sea. Given the different eastward and westward years chosen for composite analysis for SMOS and SMAP, the agreement in eastward/westward SSS composites between JPL SMAP and LOCEAN SMOS products solidifies this finding.

The composite analysis highlights the dominance of the zonal wind over the meridional wind in driving SSS patterns. Within regions with particularly shallow shelf bathymetry, such as in the South Laptev Sea, the Ekman current has been suggested to almost completely align with wind direction or to be transported ~60 ° to the right (Dmitrenko et al., 2005; Kubryakov et al., 2016; Zatsepin et al., 2015). The strong dominance of the zonal over meridional wind component observed here and the strong North/South dipole observed in SSS composite difference plots are consistent with this suggestion that the full Ekman spiral doesn't manifest.

Meridional wind stress also does appear to play a role in plume transport but only in the absence of strong zonal wind stress. This has previously been shown to be true for both 2014 (Janout et al., 2020) and 2018 (Tarasenko et al., 2021), where the wind is primarily north-westward and fresh water is transported directly offshore. Both LOCEAN SMOS and JPL SMAP support this.

There has historically been some debate as to the role of the AOI in controlling SSS variability, both locally (Bauch et al., 2010; Janout et al., 2015; Steele and Ermold, 2004), and on a full Arctic basin scale (Morison et al., 2012; Rabe et al., 2014). The mean eastward zonal surface stress in this region is found to be correlated to the mean AOI in June-August over 1993-2022 (r = 0.49), and this correlation is particularly strong over the satellite period (2010-2022) (r > 0.64).

Very similar spatial patterns are found when calculating composites from the three years of maximum and minimum (June-September) AOI as when calculating composites from years of maximum and minimum (June-August) ERA5 zonal surface stress (not shown). The similar spatial patterns highlight that local wind variability in this region is predominantly governed by large-scale dynamics over this period. The considerable variability in correlation strength (depending on time period analysed) suggests there may be some decadal variability in the extent to which the AOI controls local wind forcing in this region. In addition, the decline in summer sea ice will increase the area of atmospheric influence and in turn could alter how strongly coupled the AOI is to local wind forcing in this region.

### 4.3 Vertical distribution of plume

Nearshore in-situ data suggests that the two modes of SSS variability, visible under eastward/westward wind forcing appear to be related to very different stratification dynamics (Figure 3). In 2016, in-situ and GLORYS12V1 SSS agree particularly well and show a well-mixed very fresh plume nearshore (Figure 3), likely driven by the strong consistent onshore Ekman transport driving downwelling (Osadchiev et al., 2021). This year (2016) stood out as having a particularly well mixed plume compared to all other in-situ data in this region, the extent of which had not previously been observed (Janout et al., 2020) . A similar dynamic appears to be visible in 1994, where strong eastward wind stress is coincident with a coastally confined and well-mixed plume (not shown but visible in in-situ data and GLORYS12V1 SSS and SST).

Conversely, in-situ data showed a strongly stratified fresh layer in 2008, 2011 and 2019, even in shallow regions on the shelf (Osadchiev et al., 2021), which is poorly represented nearshore in GLORYS12V1 (Figure 3). The strong stratification on the shelf, visible in in-situ data in these years, suggests that the fresh layer is more strongly stratified in years with considerable northward spreading. This phenomenon appeared true in 1994 and 2016, where strong onshore Ekman transport appeared to drive the well-mixed plume observed. Hence, despite that the shallow shelf is below the calculated Ekman depth for this region

(37 m) (Baumann et al., 2018; Tarasenko et al., 2021), Ekman transport plays a role in controlling vertical stratification, at least in years where eastward wind stress drives onshore transport and mixing / downwelling (Lentz and Helfrich, 2002). It is also possible that the magnitude of river discharge is a dominant control on the vertical distribution of SSS, given there is no conclusive evidence that the surface fresh layer varies with cumulative runoff.

This hypothesis was not tested as the constant well-mixed plume nearshore suggests GLORYS12V1 is not capable of fully representing plume stratification dynamics in this complex environment. Other model output was considered for use (including CMEMS TOPAZ, GLORYS2V4, ORAS5, GloSea5/FOAM, CGLORS), but all models considered show the shallow shelf to be well-mixed in all years considered. The challenge of accurately representing mixing/stratification dynamics in Arctic shallow shelf seas has been widely documented (Janout et al., 2020; Hordoir et al., 2022). Given all models used here have many vertical levels but are all (except CMEMS TOPAZ) on z-level grids, it is likely the overmixing issue is a result of z-level vertical grids, as previously suggested (Arpaia et al., 2023; Wise et al., 2022; Heuzé et al., 2023). Even in years with a mostly well-mixed plume (EG 2016), in-situ data typically shows a more saline layer at depth in certain regions on the shelf, which is almost never captured by GLORYS12V1. The challenge of accurately modelling stratification in Arctic shallow shelf seas and the very limited availability of in-situ data on the shelf prevents a more in-depth analysis of the representation of vertical plume structure within GLORYS12V1. It may be useful for future studies to consider if the inclusion of interannual runoff forcing would improve representation of stratification dynamics.

## 4.4 Sea surface temperature / sea ice concentration variability

SST is known to be a useful indicator of plume location in this region (Dmitrenko et al., 2005; Osadchiev et al., 2021; Tarasenko et al., 2021). During the summer, Lena River water is typically at around 16 °C before entering the Laptev Sea, which is much warmer than the typical SST below sea ice of below 0 °C (Juhls et al., 2020). This sets up the gradient in SST that is present over the Laptev Sea (Appendix A Figure A2), with temperatures below 0 °C off the continental shelf and below sea ice and temperatures above 4 °C present over much of the shelf. Similar results have previously been shown from in-situ data, with offshore SSTs typically below 0 °C and SSTs near the mouth of the Lena River typically over 3 °C and up to 10 °C in the last 2 decades (Osadchiev et al., 2021). This represents a significant increase in September near-shore SSTs over the last several decades (Kraineva and Golubeva, 2022; Polyakov et al., 2005).

Many studies have considered the dominant drivers of SSS interannual variability and of the seasonal and decadal variability in SST (Janout et al., 2020; Osadchiev et al., 2021), but few have considered whether SSS and SST co-vary with distance from the mouth of the Lena and in turn what drives interannual variability in SST in this region. The lagged-correlation and composite analyses shows that zonal wind component is a key driver of interannual variability in SST as well as of SSS. This finding highlights that correspondence between SSS and SST is not only driven by their common source but also by their

common driver of interannual variability. The strong correspondence between eastward/westward SSS and SST composites on the shallow Laptev shelf is unsurprising given that warm and fresh Lena River water dominates oceanic properties in this region.

Whilst the eastward/westward composites appear similar, considerable differences are observed between the SSS and SST composite difference and correlation plots. The SSS composite (eastward-westward) difference plots suggest a North/South dipole where eastward forcing appears to drive onshore / south-eastward transport of fresh SSS anomalies and westward wind forcing drives offshore / northward transport of fresh SSS anomalies. In the lagged-correlation analysis between mean eastward turbulent surface stress and SSS, this pattern is highlighted by the dipole between the strong negative correlation nearshore

and strong positive correlation at the edge of the continental shelf. Conversely, the SST composite (eastward-westward) difference plots show an East/West dipole where eastward surface stress drives eastward transport of warm SST anomalies and westward surface stress drives north-westward transport of warm SST anomalies. The lagged-correlation analysis between June to August eastward turbulent surface stress and SST consists predominantly of a strong negative correlation in the northern Laptev Sea. Whilst there is a weak region of positive correlation in the East Siberian Sea, which would create the

East/West dipole described above, it is not significant. The difference in strength of correlation indicates that whilst westward wind stress drives a strong increase in SST (and/or eastward wind stress a strong decrease in SST) in the northern Laptev Sea, eastward/westward wind stress drives a much smaller change in SST in the East Siberian Sea. These differences in composite difference plots likely occur due to feedback cycles between SST, SIC, SSS and albedo.

Hence, whilst the zonal wind plays a key role in controlling both SSS and SST patterns, the differences between SSS and SST composite difference and correlation plots highlight that this warm and fresh water is exposed to very different thermal and freshwater forcing after entering the Laptev Sea. Comparing the responses of SSS and SST provides unique insight into understanding the contribution of the zonal wind in distributing warm riverine anomalies and the contribution of summer heating to the September SST pattern.


Regardless of differences in SSS and SST composite difference plots, the zonal wind clearly controls plume propagation. Under eastward wind forcing, it transports the fresh, warm plume along the coast to the East Siberian Sea, and otherwise, under westward wind forcing, it transports the plume offshore to the Northern Laptev Sea.

Zonal wind forcing is also a dominant control on the spatial distribution of September SIC in this region. The similarity in correlation patterns between eastward wind stress and GLORYS12V1 SST and SIC and the strong correlation between mean SST over the Laptev Sea and September SIC highlight the strongly coupled nature of SST and SIC in this region. The strong correlation previously found between river-water fraction and melt-water fraction suggests that early plume transport may drive sea ice melt in that region (Bauch et al., 2013). However, despite this strong correspondence, the initial heat brought by

river runoff is only suggested to contribute ~10% to sea ice breakup in early spring (Dean et al., 1994). However, the initial loss of sea ice near the river mouth and the dark-coloured water that replaces it (high in dissolved and suspended particulate matter) alters surface albedo and increases heat absorption creating a strong positive feedback (Bauch et al., 2013; Park et al., 2020). As SSTs are cooler than atmospheric air temperature in summer, SSTs will continue to warm until atmospheric temperatures start to cool in autumn (Janout et al., 2016). The strongly stratified summer halocline also increases stability of

the water column, making summer heating more effective (Osadchiev et al., 2021). Whilst warm summer air temperatures will drive a warming of SST in open water regions, freshwater input from precipitation has a negligible impact on SSS and sea ice melt only plays a small role in altering summer SSS (Dubinina et al., 2017). These differences drive the observed differences in composite difference and correlation plots. The SSS composite difference plots represent just the direct response of SSS to the zonal wind (offshore/nearshore). The SST composite difference plots also highlight the importance of the SST/SIC positive

feedback whereby warm river runoff drives sea ice melt, in turn increasing the area of shallow open water exposed to the warm atmosphere, and further driving SST warming in newly open water regions. The much stronger correlation between eastward wind stress and SST and SIC in the northern Laptev Sea, compared to the East Siberian Sea, may be related to this SST/SIC positive feedback and the timing and/or region of sea ice retreat. It is possible that offshore transport (driven by westward winds) drives earlier and/or more expansive sea ice melt, which would in turn alter the area of open water exposed to the

atmosphere and the length of time it is exposed to the warm summer atmosphere, in turn driving more dramatic warming. Whilst this hypothesis is consistent with results here, further work would be needed to confirm this. It is worth noting that the similarity between SSS and SST eastward/westward composites highlights the importance of the zonal wind in modulating this SST/SIC warming positive feedback.

The difference in spatial pattern of SIC under eastward and westward wind forcing and the relationship between SST and SIC suggests zonal wind is not only a key driver of variability in SSS and SST but also of Laptev SIC. There have previously been Arctic-wise studies that have suggested that the summer AOI is a good predictor of September SIC (Ogi et al., 2016), but the same has not yet been suggested locally in the Laptev Sea. The consistency of SST composites calculated in this study from years of strong eastward (/westward) turbulent surface stress with that of strongly positive (/negative) AOI years supports that

large-scale circulation appears to be the dominant driver of variability in this region.

Previous work in this region has also suggested that variability in SSS is unrelated to sea ice dynamics (Osadchiev et al., 2021). However, both the composite and correlation analysis here show that variability in zonal wind stress does play a role in controlling SST and SIC. Attributing variability in SST and in turn SIC to zonal wind stress is complex due to the SST/SIC

warming positive feedback described above and the strong decline in SIC visible in the Laptev (Kraineva and Golubeva, 2022).

However, the spatial pattern of GLORYS12V1 eastward and westward SST composites is consistent regardless of time period chosen (the full GLORYS12V1 time period, the LOCEAN SMOS time period or the JPL SMAP time period), suggesting this

relationship does not only exist due to the SIC trend (i.e. if years of westward/eastward forcing are present earlier/later in the timeseries). In addition, the spatial pattern of variability visible in both SST composite difference plots and in eastward turbulent surface stress and SIC correlation plots is different from the long-term pattern of SST warming or SIC decline (between 1993-2002 and 2010-2019 in GLORYS12V1), which suggests a pattern of more rapid warming distributed across the continental shelf. The consistency of SST composites shown, the difference in spatial pattern of SST under differing wind forcing and the strength of correlation and similarity in correlation pattern between eastward turbulent surface stress and SST and SIC support that wind stress is a control on SST and in turn September SIC. Further work is needed to investigate if variability in SSS and SST impact later sea ice formation as well as September SIC.

## 4.5 Implications with climate change

The increase in riverine heat has already contributed to a regional loss of sea ice, and it has been suggested that warming river discharge is a key control on basin-wide SIC (Dong et al., 2022; Park et al., 2020). It is also clear that the increase in river runoff will increase the freshwater content of the Laptev Sea and have implications for local and Arctic-wide stratification dynamics as well as for local biogeochemistry. However, the weak correlations with spring, summer and annual runoff and dominance of zonal wind as a key driver of SSS and SST interannual variability suggests that understanding variability in wind stress and if it is likely to change is the key to predicting future freshwater transport from the Eurasian shelf seas.

This is all the more relevant as the dominance of wind stress variability is only likely to increase with the loss of sea ice cover. Prior to the mid-2000s, the Lena plume typically remained strongly-stratified and confined to the Laptev Sea shelf, constrained by the extensive sea ice cover and small region of atmospheric influence (Janout et al., 2020). The loss of sea ice cover in the Laptev Sea is enlarging the area in contact with the atmosphere and increasing the time of atmosphere-ocean exposure. The strong influence of the AOI on local wind stress in this region, and the increase in correlation strength over the more recent time period, highlights the need to investigate how large-scale atmospheric circulation will change over the Arctic to understand future changes in Laptev Sea freshwater transport. This relationship is only likely to become stronger given the AOI is suggested to have increased in variability in recent decades (Armitage et al., 2018; Morison et al., 2021), and as future sea ice loss will only strengthen coupling between large-scale and local wind dynamics. These changes have already and will likely continue to expand the region of potential riverine freshwater influence (Janout et al., 2020; Johnson and Polyakov, 2001; Zhuk and Kubryakov, 2021) and in turn have the potential to speed up transport between the shelf seas and central Arctic (Charette et al., 2020).

However, the impact this will have on the wider Arctic will strongly depend on changes in stratification dynamics in the Laptev Sea. Whilst it is likely that stratification dynamics will change as the region of potential freshwater influence expands, it remains uncertain what the dominant drivers of this change will be and in turn how this change will manifest. On the one hand,

having a larger open water region exposed to wind-driven mixing for longer periods could deepen stratification, increasing the tendency of a well-mixed plume (Janout et al., 2020). This appeared to occur in 2016 and seems likely under strong eastward wind forcing, where the fresh water is transported eastwards, driving downwelling and mixing and creating a coastally confined well-mixed plume. Alternatively, the increase in river runoff to the Arctic could strengthen surface stratification (Nicolì et al., 2020; Nummelin et al., 2016) and increase the likelihood of a very shallow plume that extends out northwards towards the central Arctic. It is also possible that the likelihood of both of these alternating states could become more frequent, with the increased influence of wind variability with the loss of sea ice cover (Janout and Lenn, 2014). Changes in stratification will be strongly coupled to changes in sea ice dynamics, not only in summer but also year round, and will have implications for the timing, magnitude and region of water mass formation / transformation in the Laptev (Preußer et al., 2019). Untangling all these compounding changes remains a challenge and will only be solved by a unified approach bringing together a combination of different data products and types including in-situ data, satellite data and model output. The long satellite SSS timeseries has, and with the launch of the Copernicus Imaging Microwave Radiometer (CIMR) will continue to be, a valuable asset in understanding Arctic wide freshwater transport. Understanding these processes will be further aided by the launch of higher resolution satellites for mapping sea surface geostrophic (and total) velocity, including the Surface Water and Ocean Topography (SWOT), SeaSTAR, Harmony and ODYSEA (Gommenginger et al., 2019; Morrow et al., 2019; Suess et al., 2022; Lee et al., 2023).

## 5 Conclusions

Satellite SSS agrees well with in-situ data ($r \geq 0.84$) and provides notable improvement compared to GLORYS12V1 SSS ($r \leq 0.80$) and the other reanalysis products ($r \leq 0.83$) considered in capturing patterns and variability observed by in-situ SSS data. Hence, satellite SSS provides a useful tool to strengthen our current understanding of Laptev Sea and wider Arctic SSS dynamics, particularly in regions with strong SSS gradients. Comparison between satellite and in-situ data in this region highlights the need for more near-surface in-situ data for validation in this region, particularly nearshore over the lowest salinities. The current lack of nearshore low salinity in-situ data limits the confidence in and ability to validate satellite data over regions of very low salinities (< 20 pss) and limits our understanding of vertical stratification over the shelf, particularly given its high spatial and temporal variability.

GLORYS12V1 and satellite SSS data confirms what in-situ data has previously suggested: that the zonal wind is the dominant driver of offshore/onshore Lena River plume transport, with strong consensus in SSS patterns under eastward and westward wind regimes in GLORYS12V1, LOCEAN SMOS and JPL SMAP. Annual, summer and spring runoff do not appear to play a role in controlling interannual variability in SSS, SST or SIC in the Laptev and East Siberian seas. The zonal wind also plays a key role in driving SST variability and appears to drive spatial variability in SIC across the Laptev and East Siberian Seas. The differences in spatial patterns of SSS and SST under eastward/westward wind forcing highlight the importance of the

zonal wind for dispersing riverine heat and in turn controlling the SST/SIC positive feedback, which plays a considerable role in driving further SST warming in shallow open water regions. The dominance of local wind stress as a driver of salinity and temperature variability, and its strong correlation with the AOI and large-scale atmospheric circulation, highlights the need to understand how local and large-scale wind stress has and will change as the Arctic warms in order to predict changes in freshwater storage and transport from the Eurasian shelf seas. The interconnected nature of SSS, SST and SIC in this region highlights the challenge but also the need to understand this region as a system rather than trying to understand drivers of individual components in isolation. This will prove vital to be able to predict how the conflicting changes in this region will impact both this region and wider Arctic sea ice dynamics and freshwater transport.

**Appendix A**

Only observations in the upper 10m are used for comparison with satellite data (Figure A1). The same analysis was conducted using only data in the upper 5m with no significant improvement. The analysis shown here is for the upper 10m to retain as much data as possible.

**Table A1: Cruises, vessels and time-periods of salinity and temperature in-situ data used for analysis of vertical profiles and comparison with satellite data**

| Cruise Name | Vessel | Time Period | Reference |
|---|---|---|---|
| UDASH dataset (incl NABOS cruises 2013, 2015) | Numerous | 2010-2015 | (Behrendt et al., 2017) |
| NABOS cruise 2018 UCTD | Akademik Tryoshnikov | 3rd -17th October 2018 | (Janout et al., 2019) |
| | Akademik Lavrentyev | 20th September – 20th October 2016 | Supplementary materials (Osadchiev et al., 2021) |
| | Akademik Mstislav Keldysh | 23rd September – 13th October 2019 | |

All satellite and reanalysis products described above are compared with in-situ data over 2015-2020. The regridded SMOS data and GLORYS12V1 reanalysis (on a 0.25 ° grid) are used for comparison with in-situ data. Both Pearson correlation coefficients and root-mean square difference (RMSD) values are calculated for each individual product at all collocations (across the entire area and time period) between in-situ data and that product. Correlation coefficients and RMSD values are also calculated only where all products have a collocation with in-situ data. However, over 2015-2020, few in-situ observations are collected sufficiently near the surface (< 10 m) over regions where all satellite products obtain an SSS measurement (only 37 collocations). Therefore, RMSDs and correlation coefficients are also calculated for SMOS products and reanalyses over the longer SMOS time period (2011-2020) to obtain more collocations (228). JPL SMAP and LOCEAN SMOS have particularly high correlation coefficients and low RMSD values and agree well so are used for further analysis.

# In-situ observations 2010-2020

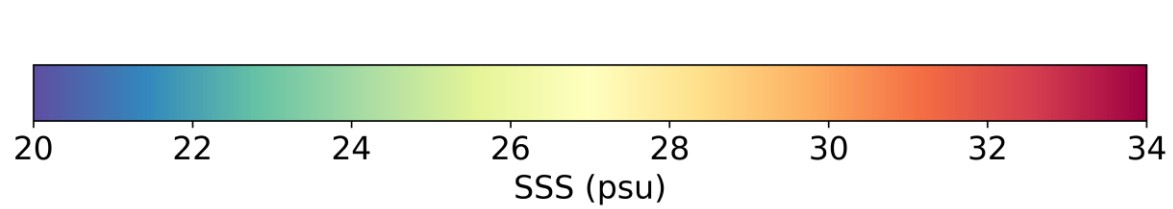

**Figure A1: In-situ data (<10 m) used for validation of satellite and reanalysis products, coloured by their salinity value. Data with black circles were collected over the SMAP period (2015-present), and those without black circles were collected over the SMOS period (2010-present).**

**Table A2: Correlation coefficients from in-situ SSS data < 10m over 2015-2020 (left) and 2010-2020 (right) with GLORYS12V1, BEC SMOS and LOCEAN SMOS products regridded at a 0.25 degree spatial resolution, JPL SMAP in regions where the provided SSS uncertainty is less than 1, RSS SMAP and the four CMEMS global ensemble reanalysis products: GLORYS2V4, ORAS5, GloSea5, and C-GLORS05. Correlation coefficients are calculated both at all points where an individual product is collocated with in-situ data (All obsv <10m) and for only where all products had a collocation point near in-situ data (Common obsv <10m). There are 57 collocations between all products over 2015-2020 and 377 collocations over 2010-2020. The p values associated with correlation coefficients are not included but are all << 0.01.**

| | 2015-2020 | | | | 2010-2020 | | | |
|---|---|---|---|---|---|---|---|---|
| | All obsv <10m | | Common obsv <10m | | All obsv <10m | | Common obsv <10m | |
| | Num obsv | Corr coeff | Num obsv | Corr coeff | Num obsv | Corr coeff | Num obsv | Corr coeff |
| GLORYS12V1 regridded onto 0.25° grid | 222 | 0.80 | 57 | 0.75 | 1667 | 0.78 | 377 | 0.65 |
| BEC SMOS regridded onto 0.25° grid | 133 | 0.79 | | 0.79 | 396 | 0.76 | | 0.75 |
| LOCEAN SMOS regridded onto 0.25° grid | 132 | 0.86 | | 0.92 | 406 | 0.84 | | 0.84 |
| JPL SMAP (where uncertainty < 1) | 100 | 0.92 | | 0.95 | | | | |
| RSS SMAP | 67 | 0.93 | | 0.93 | | | | |
| C-GLORS05 | 219 | 0.75 | | 0.73 | 1672 | 0.72 | 377 | 0.61 |
| GloSea5 | 219 | 0.84 | | 0.78 | 1672 | 0.88 | | 0.75 |
| GLORYS2V4 | 219 | 0.81 | | 0.75 | 1672 | 0.72 | | 0.48 |
| ORAS5 | 219 | 0.85 | | 0.83 | 1672 | 0.89 | | 0.80 |

**Table A3: Root mean square differences (RMSD) from in-situ SSS data < 10m over 2015-2020 (left) and 2010-2020 (right) with GLORYS12V1, BEC SMOS and LOCEAN SMOS products regridded at a 0.25 degree spatial resolution, JPL SMAP in regions where the provided SSS uncertainty is less than 1, RSS SMAP and the four CMEMS global ensemble reanalysis products: GLORYS2V4, ORAS5, GloSea5, and C-GLORS05. RMSDs are calculated both at all points where an individual product is collocated with in-situ data (All obsv <10m) and for only where all products had a collocation point near in-situ data (Common obsv <10m). There are 57 collocations between all products over 2015-2020 and 377 collocations over 2010-2020.**

| | 2015-2020 | | | | 2010-2020 | | | |
| | All obsv <10m | | Common obsv <10m | | All obsv <10m | | Common obsv <10m | |
| | Num obsv | RMSD | Num obsv | RMSD | Num obsv | RMSD | Num obsv | RMSD |
|---|---|---|---|---|---|---|---|---|
| GLORYS12V1 regridded onto 0.25° grid | 222 | 3.16 | 57 | 4.28 | 1667 | 1.88 | 377 | 2.69 |
| BEC SMOS regridded onto 0.25° grid | 133 | 2.90 | | 3.74 | 396 | 2.21 | | 2.25 |
| LOCEAN SMOS regridded onto 0.25° grid | 132 | 2.53 | | 2.74 | 406 | 2.07 | | 1.97 |
| JPL SMAP (where uncertainty < 1) | 100 | 1.85 | | 2.19 | | | | |
| RSS SMAP | 67 | 2.77 | | 2.16 | | | | |
| C-GLORS05 | 219 | 3.75 | | 4.18 | 1672 | 2.30 | 377 | 2.67 |
| GloSea5 | 219 | 2.83 | | 3.88 | 1672 | 1.40 | | 2.40 |
| GLORYS2V4 | 219 | 3.04 | | 3.93 | 1672 | 2.14 | | 3.25 |
| ORAS5 | 219 | 2.75 | | 3.74 | 1672 | 1.41 | | 2.42 |

The satellites products show a good agreement with in-situ measurements within the top 10m, with a correlation coefficient typically higher than 0.62 and up to 0.83. The RMSD with in-situ is typically between 1.1 and 1.65. Despite this relatively high error in RMSD, due to the large range of SSS observed over this small area (5 to 35), both datasets are well correlated. JPL SMAP, LOCEAN SMOS, and the median sat product stand out as having particularly high correlation (r ~ 0.8) coefficients compared to all other products. Over the full SMOS period, the LOCEAN product correlates strongly with in-situ data (r = 0.83) but the BEC product is less strongly correlated (r = 0.67).

The collocated in-situ data (common obsv <10m) are all located in low sea ice regions (< 30% SIC), where satellite SSS retrievals are possible. Over the Laptev Sea, the strong horizontal gradient in SSS maintains lower salinities nearshore on the continental shelf and relatively higher salinities > 30 offshore. Therefore, the salinity range captured by in-situ observations only collocated with one satellite product/GLORYS12V1 typically includes a larger range of salinities (with more SSS values <30) than that captured by in-situ observations collocated with all products. Hence, the correlation coefficients of almost all products are larger when considering all in-situ observations collocated with that product due to the larger range in SSS than when considering only in-situ observations collocated with all products.

Whilst GLORYS12V1 appears to correlate well with in-situ data when considering all its collocations (r > 0.79 over 2015-2020 and r > 0.78 over 2011-2020), the correlation deteriorates when only considering observations where all satellite products have a collocation (r < 0.35 over 2015-2020 and r < 0.63 over 2011-2020). This same pattern is visible in all other reanalysis products considered. This decrease in correlation indicates that the reanalyses manage to replicate the large-scale horizontal gradient in SSS (between the fresh plume on the shelf and the more saline water that sits off the shelf, under sea ice) but are not capable of representing the spatial variability at lower SSS values and hence of finer scale river plume dynamics. Reanalysis RMSDs from in-situ data are also all larger than those of any satellite product. The lower RMSDs and stronger correlation coefficients of all satellite products compared to reanalyses highlight the value satellite SSS products bring to Arctic-based process studies.

**Appendix B**

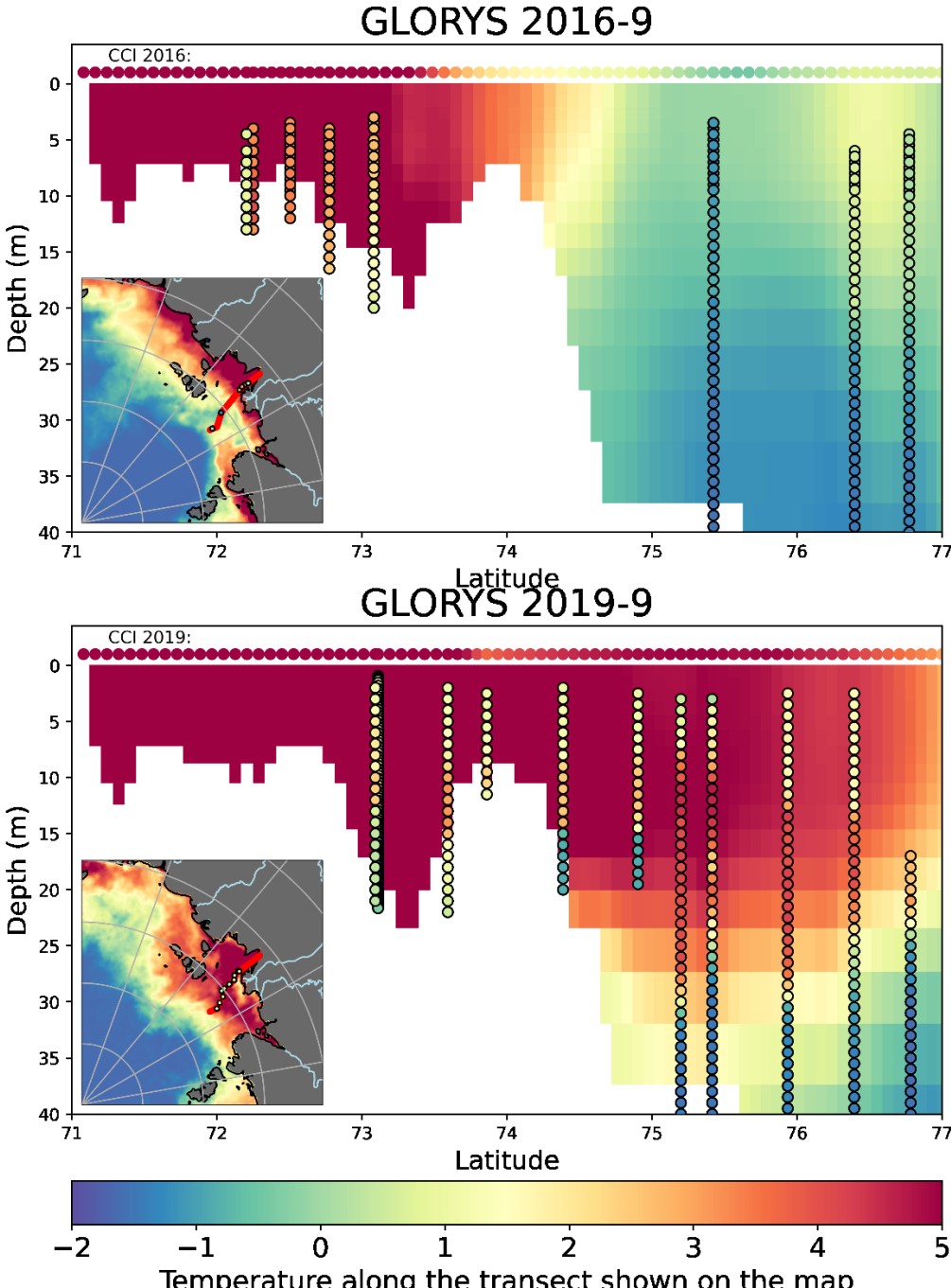

790

**Figure A2**: *GLORYS12V1 SST vertical transect in 2016 (top) and 2019 (bottom) along red transect interpolated through in-situ data (shown in map of CCI SST in bottom left for each year) with in-situ data overlaid with black rings and satellite data for that transect in CCI SST shown as a line of points.*

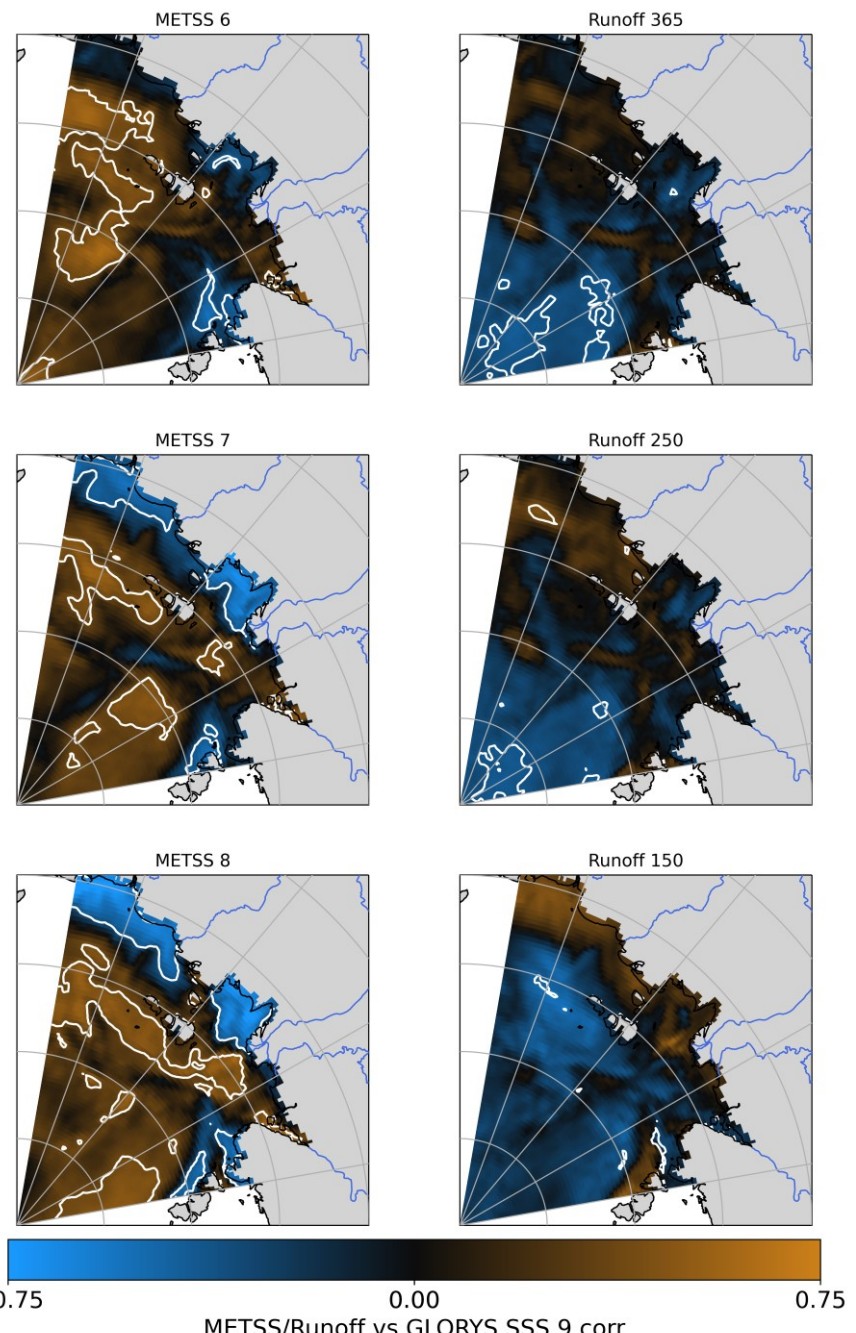

**Figure A3: Correlation between GLORYS12V1 September SSS and mean eastward turbulent surface stress (METSS) over 70-80 North and 120-160 East in June (6), July (7), August (8) (left column) over 1993-2022. Correlation between GLORYS12V1 September SSS and cumulative Lena River runoff over the full year (Julian day 365), in summer (Julian day 250) and in spring (Julian day 150) (right column) over 1993-2022. Regions where correlations are statistically significant (p≤0.05) are denoted by the white contour and brighter colours.**

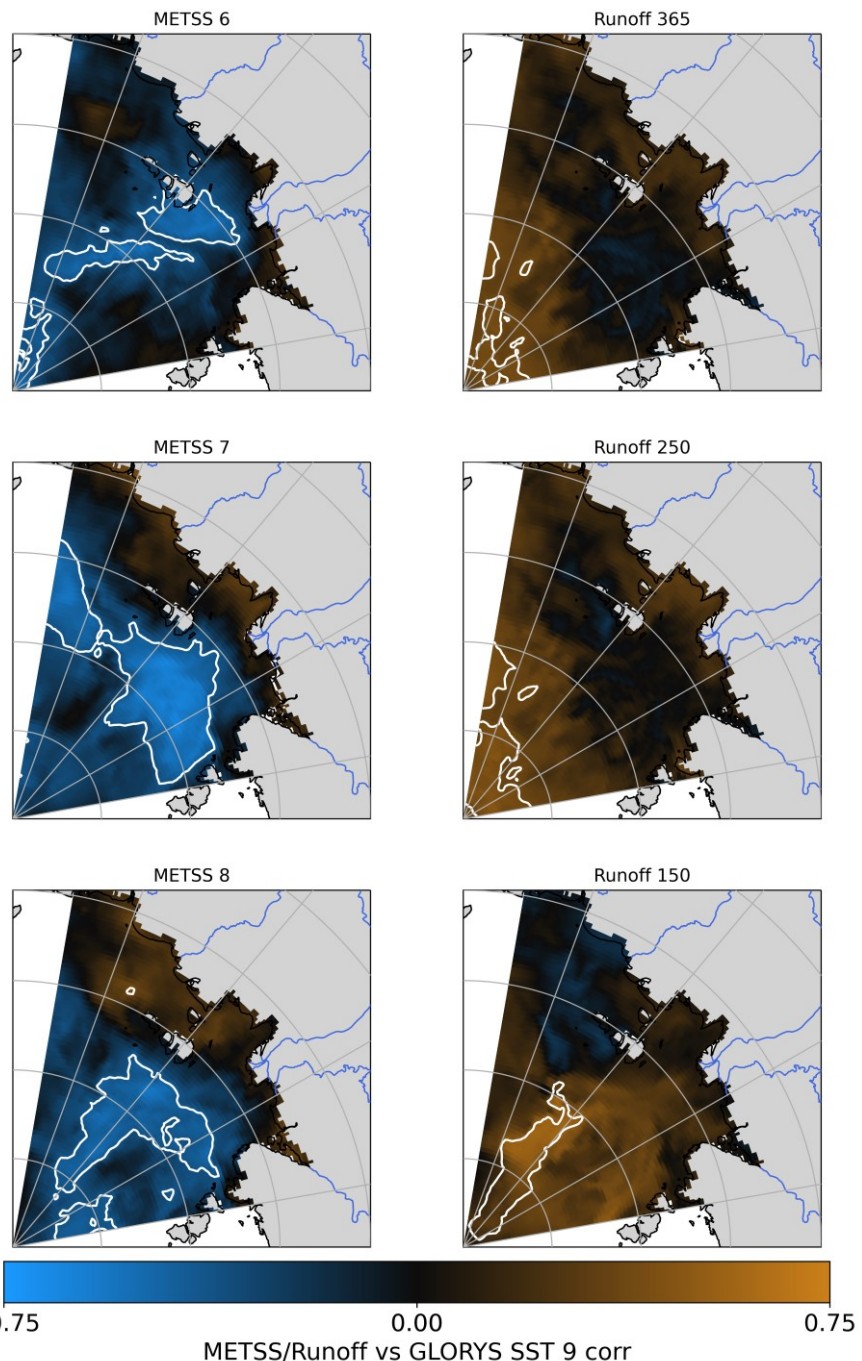

Figure A4: Correlation between GLORYS12V1 September SST and mean eastward turbulent surface stress (METSS) over 70-80 North and 120-160 East in June (6), July (7), August (8) (left column) over 1993-2022. Correlation between GLORYS12V1 September SST and cumulative Lena River runoff over the full year (Julian day 365), in summer (Julian day 250) and in spring (Julian day 150) (right column) over 1993-2022. Regions where correlations are statistically significant (p≤0.05) are denoted by the white contour and brighter colours.

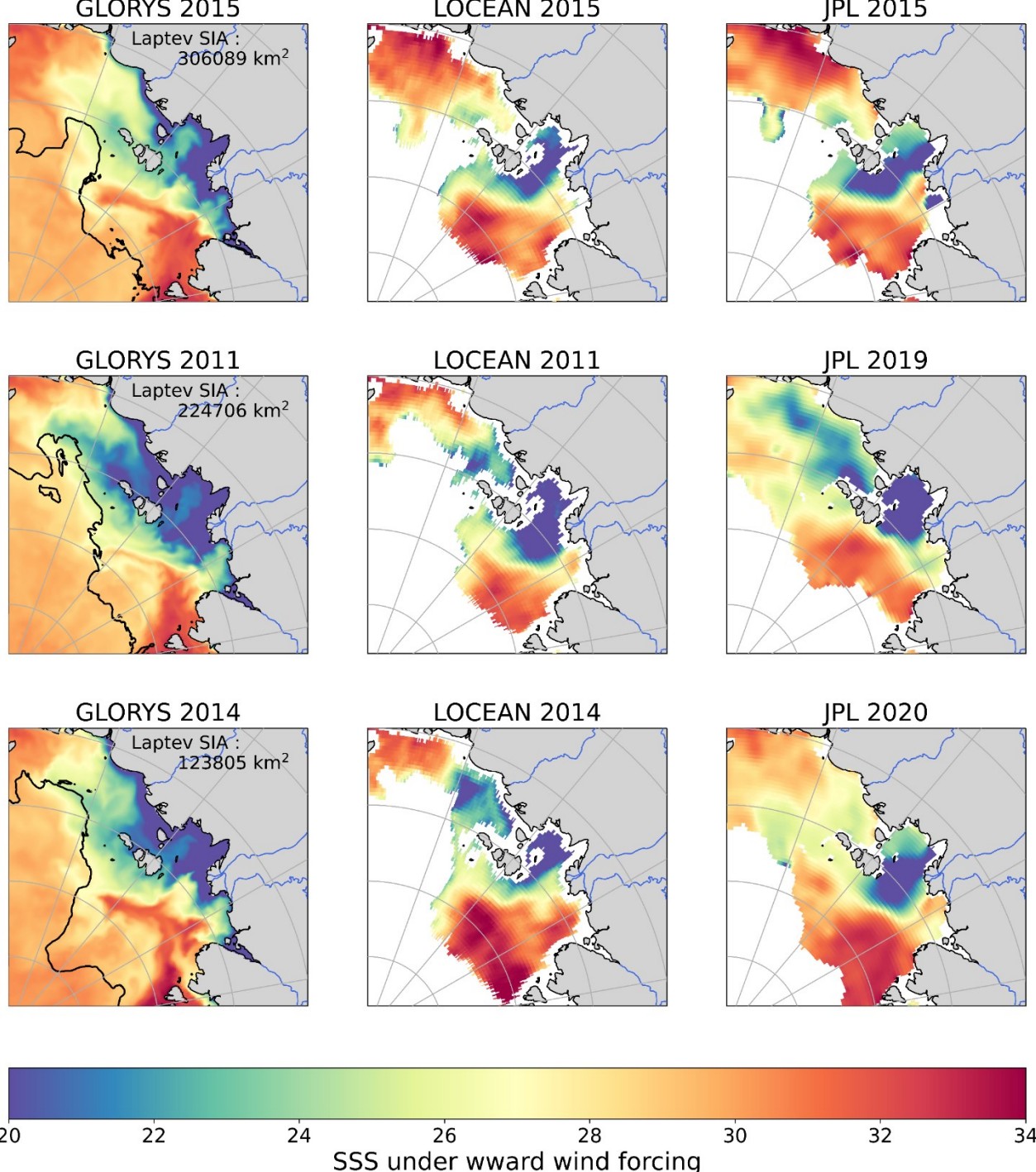

**Figure A5:** *Years of westward wind forcing for all years used to calculate westward composites for (left to right) GLORYS12V1 SSS and LOCEAN SMOS (2019, 2011, 2013) and for JPL SMAP (2019, 2015, 2020). The GLORYS12V1 mean 30% sea ice concentration contour and mean GLORYS12V1 sea ice area (SIA) in the Laptev Sea (120-140, 68-85N) for each year shown is overlaid on that year's plot.*

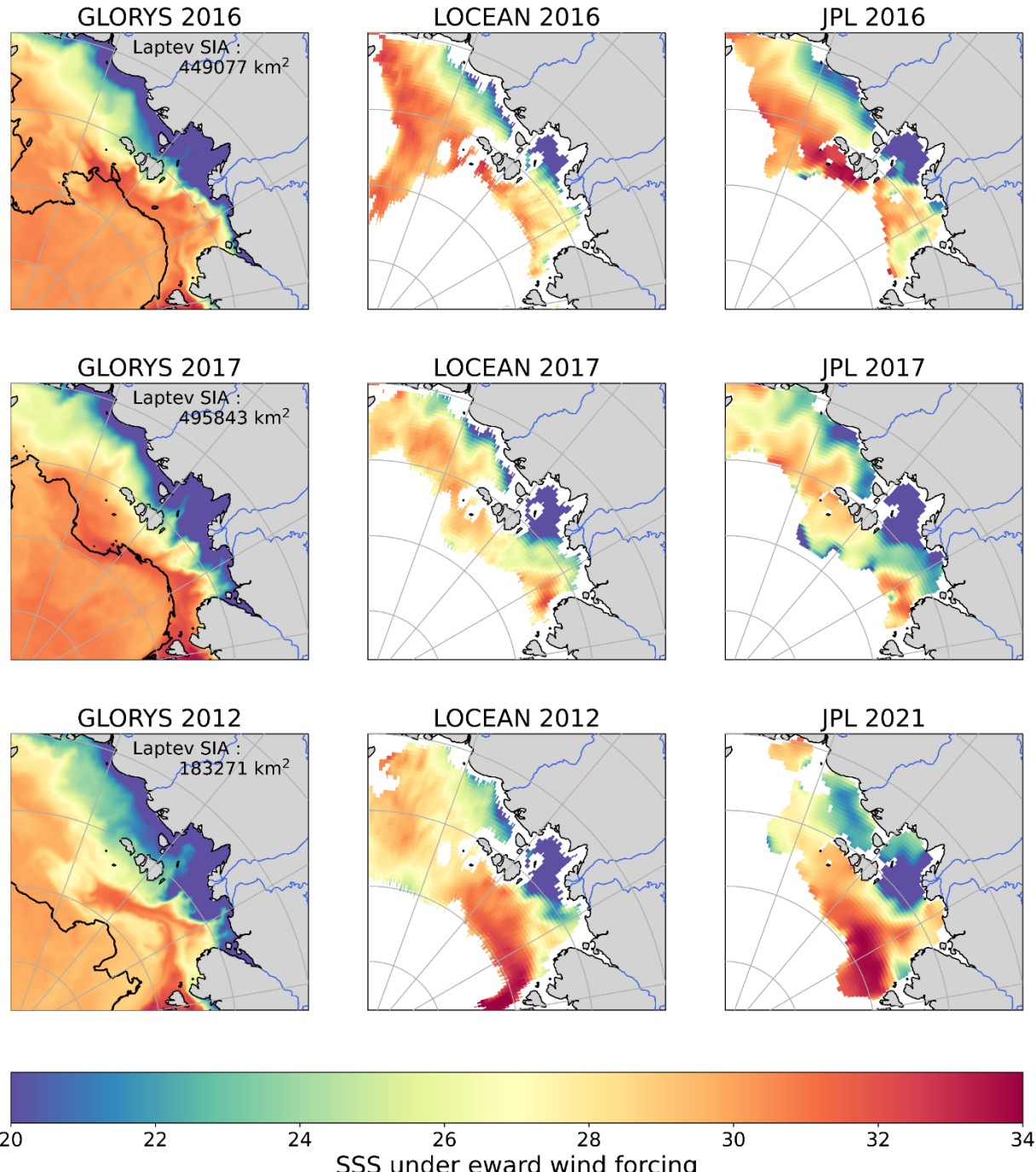

**Figure A6: Years of eastward wind forcing for all years used to calculate eastward composites for (left to right) GLORYS12V1 SSS and LOCEAN SMOS (2016, 2017, 2012) and for JPL SMAP (2016, 2017, 2021). The GLORYS12V1 mean 30% sea ice concentration contour and mean GLORYS12V1 sea ice area (SIA) in the Laptev Sea (120-140, 68-85N) for each year shown is overlaid on that year's plot.**

**Data availability**

All data used in this study is open access. JPL and RSS SMAP SSS data can be obtained from the Podaac data portal. LOCEAN
SMOS data is available on the CATDS portal and SMOS BEC product is available on the BEC (Barcelona Expert Center) web page. CCI satellite SST data are available from Sea Surface Temperature Data (surftemp.net). All reanalysis products are available through the CMEMS portal.

In-situ data from the UDASH database and from Bjork (2017) are accessible on the Pangea portal. In-situ data from the NABOS cruises are available from the Arctic Data Center. In-situ data from cruises in 2016 and 2019 can be found in the supplementary materials of (Osadchiev et al., 2021).

**Author contribution**

Phoebe Hudson: Conceptualisation, Methodology, Validation, Formal Analysis, Visualisation, Writing –original draft, review & editing.

Adrien Martin: Supervision, Funding acquisition, Conceptualisation, Analysis, Writing – review & editing.

Simon Josey: Supervision, Analysis, Writing – review & editing.

Alice Marzocchi: Supervision, Analysis, Writing – review & editing.

Athanasios Angeloudis: Supervision, Analysis, Writing – review & editing.

**Competing interests**

The authors declare no conflict of interest, either financial or personal, that may have influenced the work reported here.

**Acknowledgements**

PAH was supported by the Natural Environment Research Council (NERC) SENSE Centre for Doctoral Training (NE/T00939X/1).

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
