# Peer review of "Drivers of Laptev Sea interannual variability in salinity and temperature"

_EGUsphere, 2023_

## Author Comment (AC1)

Hi Phoebe,

You're not going to like what comes next. Sorry.

We thank you for your honest and constructive review and believe it will notably strengthen the paper.

**Major comments**

I. The objective of this manuscript is unclear

Although both the abstract and the introduction are very long, they are also unfocused, so it is unclear whether the aim of this article is to:

- combine satellite, reanalysis, and in-situ data to study the plume;
- demonstrate that satellite data can be used to look at sea surface salinity in the region;
- or determine which reanalysis product is most adapted for this study.

These different objectives would then result in different structures of the manuscript, with the majority of the product comparisons and validations going to the appendix. It would also affect the time period considered. The choice of in-situ data would then be affected as well; see my next point.

The main objective of the paper: "combining satellite, reanalysis, and in-situ data to study the plume" will be clarified and the paper will be restructured to reflect this. The validation component of the paper will be de-emphasized and moved out of the results section.

II. The choice of data, especially their resolution, was surprising

The method section did not clarify much, so I am not sure which time period you worked with. That is, you mention that you use UDASH, but that stopped in 2015 and would not really help with SMAP. The manuscript needs an overarching table that says for all types of products over which time period and at which spatio-temporal resolution they are available (not the resolution at which you use them; their native one).

Coupled with the fact that the objective is unclear, I won't be able to give you a clear direction. But if you want to validate SSS, I would have looked for underway CTD data rather than CTD casts. It will be at approx. 10 m depth, but on most vessels (and especially so on Oden, i.e. for the SWERUS data) the upper 10 m of the CTD casts cannot be used anyway. If you want the upper water column, there might be ITPs nearby, and there should be at least one mooring, but I'm not sure of their time coverage.

The other thing that really surprised me is that you want to investigate a plume dynamics, have a 3-day product available… and downgrade it to monthly resolution. And then regularly show results in the manuscript that you explain with the poor resolution of your monthly product. Use the 3-day version.

And regardless of what you do, you need to say why you do so. Maybe you had a perfectly valid reason for using CTD casts and downgrading everything to one month, but you did not write it so the reader cannot know.

A table similar to that detailing in-situ data used will be added for the reanalysis and satellite products to clarify all products used, their spatial resolutions and time-periods.

The section detailing in-situ data used will be updated to more accurately reflect the range of in-situ data used for validation (not just CTD casts). The SWERUS (Oden) cruise CTD data will no longer used. NABOS underway data will be considered for use instead of CTD casts from the 2018 cruise. Table 1 will be updated to reflect these changes.

ITPs from the Mosaic cruise (from 2020) were considered for use but were >84N (and notably away from the shallow Laptev Shelf and the plume). The NABOS moorings (2013-2015, 2018-2021) were also considered for validation but are realistically too far off the shelf and too deep (minimum depth 30m) for validation of the main plume.

The reasoning behind the choice of data will be better clarified. The shift towards one main objective of understanding plume interannual variability, and the de-emphasis on validation should also help to clarify the choice of data.

III. Causation is not shown

This is my main issue with your manuscript. You do not demonstrate causation. You produce two composites, and declare that the variable you composited against explains the differences.

Let's start with the definition of the composites. I agree that on Figure 2, the circulation is different. However on Figure 4, the uncertainty is so large that some of the strongest years could in fact have a value with either sign. See for example 2012 and 2019. As rotation is involved, a metric based on the curl of the wind, or simply on the sea level pressure, may be more effective and robust.

Anyway, the outcome of the composites is that the SSS looks different. But so do the sea ice and the SST, which could both explain the SSS pattern, and even be responsible for the wind differences. Or maybe wind and SSS are both the result of another variable that is not included in your analysis.

What you are really showing is that the hypothesis that wind drives SSS is not incompatible with the observations. But as your analyses currently are, you do not demonstrate the causality. One option is therefore to just rephrase everything, removing all mentions of "the wind drives" and saying what I just wrote. But that's rather underwhelming a result.

Instead, you could utilise the 3-day product in its full 3-day glory. For each point, or for the overall region, do a lagged (temporal) correlation analysis of the relationship between wind (available at daily resolution; downgrade to 3-day) and the 3-day SSS from BEC. Then see which variable drives the other, based on these values. I personally would push the analysis and perform the same calculation also with the SST and the

sea ice, and have an overall result matrix that shows for each pair of variables for which lag the correlation is maximal and what that correlation value is.

To clarify, the grey overlay in Figure 4 is the maximum and minimum wind stress over the 4 month period (not the uncertainty). However, this comment encouraged us to rethink the 4-month metric we were using.

To better demonstrate causation, a lagged correlation analysis was run between ERA5 eastward turbulent surface stress in individual months and GLORYS SSS in September (see attached figure). The results of this analysis caused us to change to a 3-month metric (July-August) (rather than the 4-month metric previously being used). Given the highly variable nature of eastward wind stress in September, this helps to clarify Figure 4 by decreasing the range of wind stress overlaid in grey and supports that eastward wind stress is the (or at least a) dominant driver of SSS.

[Figure]

*Figure 1: Lagged temporal correlation between ERA5 eastward turbulent surface stress in April-September (4-9) and GLORYS September SSS*

Similar lagged-temporal correlation analysis are being conducted between eastward turbulent surface stress, SSS, SST and sea ice concentration.

**Less major comments, in order of appearance**

Salinity unit: You meant "psu"or really"pss"? Oceanographers use the absolute salinity in g/kg now.

We will alter the paper to not use any salinity unit.

We avoided the use of "psu" following the guidelines of the TEOS manual "Note that Practical Salinity is a unit-less quantity. Though sometimes convenient, it is technically incorrect to quote Practical Salinity in "psu"; rather it should be quoted as a certain Practical Salinity "on the Practical Salinity Scale PSS-78"." (Intergovernmental Oceanographic Commission et al., 2015).

Subsection 2.1.1: not detailed enough. Either there or in the introduction, you need to be more specific about which satellite measures at which band, has which footprint, repeat time, etc. Something similar to the first paragraph of section 2.1 of these people: https://tc.copernicus.org/articles/12/921/2018/tc-12-921-2018.pdf , but for all sensors (at least SMOS and SMAP, and then explanations about how the different products combine them).

The shift in objective away from the focus on validating satellite SSS data also shifts the focus away from the specific concerns/issues raised here. Nonetheless, we will consider how to include additional details in the revised manuscript where necessary and based on the updated content.

Subsection 2.2.1: also not detailed enough; what do you mean by "correlation" and "RMSD"? I assume that you took all points available, regardless of location and time, and basically did a regression? Given that the plume is both time and space dependent, as shown on your figure, I would recommend you verify the temporal and spatial accuracy separately. You may need more points for this, agreed, but see Major comment II.

The metrics calculated will be better defined in methods.

Section 3.1: You do not show the RMSD. See my previous comment anyway, but that could be added to the table -which could be shortened once the manuscript is more focused, see Major comment I.

If the RMSD values are mentioned in text, the table containing RMSD values will be included in the Appendix.

Sea ice: Line 377 onwards you give statistics of the sea ice area, without specifying over which region. Overall, and based on the figures shown in the manuscript, the area does not seem to matter as much as the southern / eastern extent. I would rather use such extent, if doing the correlations suggested above. If not, then do not even quantify it; your maps are very clear.

The region over which sea ice area is calculated is specified in the Methods section : "The GLORYS12V1 sea ice area (SIA) in September in the Laptev Sea (defined to be between 120-145 °E and 68-85 °N for the purpose of calculating SIA) is calculated from GLORYS12V1 sea ice concentration for all years used in the (eastward and westward) composite analysis. The mean "eastward" and "westward" SIA is then calculated as the mean of SIA in the three most eastward and westward years respectively."

Runoff: Line 405, the discussion starts with an analysis of the correlation between runoff and SSS. It is not specified, but I assume that the runoff values are published elsewhere? If so, reproduce them here, and do a proper (lagged) correlation analysis.

Correlations with runoff will be considered for inclusion in the next manuscript version. Interannual variability in GRO runoff values and their (cor)relation to variability in sipatial pattern of SSS was considered early on in this analysis but no strong relation was found. In addition, the GLORYS12 reanalysis is forced with monthly climatological runoff values but manages to replicate variability in the spatial pattern of the plume relatively well, suggestting interannually varying runoff is not needed to replicate the spatial pattern of plume propagation.

Arctic Oscillation: Same comment, no information about where the AOI comes from and the correlation analysis is not shown.

The source of Arctic Oscillation data will be clarified in the methods section.

References:

Intergovernmental Oceanographic Commission, Research, S. C. on O., and Oceans, I. A. for the P. S. of the: The International thermodynamic equation of seawater – 2010: calculation and use of thermodynamic properties. [includes corrections up to 31st October 2015] ., UNESCO, 2015.

---

## Author Comment (AC2)

Dear coauthors,

Your paper offers valuable insights into the drivers of Laptev Sea dynamics and interannual variability in salinity and temperature. It provides evidence that the salinity and temperature signatures agree in different reanalyses and satellite products under different wind regimes. Addressing the major comments proposed will improve the clarity and robustness of the research, enhancing its contribution to the scientific community.

We thank you for your positive and in depth review and agree that implementing suggested changes will strengthen the paper.

**Major Comments:**

***The focus of the paper:***

The stated objective of the paper is to determine the drivers of the interannual variability of the Laptev Sea dynamics. However, a significant portion of the paper is devoted to the validation/intercomparison of satellite Sea Surface Salinity (SSS) products. While the effort to validate and compare these products is commendable, it appears to dominate the narrative, diverting attention from the primary objective of identifying the drivers of Laptev Sea dynamics.

To address this concern, I see two options:

a) Dividing the current study into two separate papers, each with a distinct focus and stating clearly the objectives (Validation and Intercomparison of Satellite SSS Products / Drivers of Laptev Sea Interannual Variability in Salinity and Temperature).

b) Focus on the primary objective during the narrative and if you feel that the validation/intercomparison of satellite SSS products is an essential part of the methodology, it would be beneficial to include at least one example figure showcasing the different products. Additionally, providing information on the p-value of your correlations, bias, and spectral analysis to assess the effective resolution of the products will improve the methodological rigor and transparency of the study.

We agree that the validation distracts from the main objective. The abstract and introduction will be shortened to help more clearly highlight the focus of the paper. The section validating in-situ data will be moved to help better emphasize the main objective.

***Methods:***

1. In subsection 2.2.2, it is not clear why the analysis is not performed with all four satellite products. To ensure the robustness of the study, it would be beneficial to explain the reasons for the exclusion, if any, of certain products in the analysis.

The reasoning behind the omission of two products will be better justified earlier in the text. Primarily, the omission was based on the correlation and RMSD values calculated for each product, so moving much of the validation section out of results will allow earlier discussion of the choice of products used in the methods section.

2. The temporal resolution of satellite SSS products is a critical factor when studying the dynamics of a region like the Laptev Sea, where rapid changes can occur over short time scales. If you choose not to use 3-day or available 8-day satellite Sea Surface Salinity (SSS) products, providing a clear and well-justified argument for this decision is crucial.

The reasoning behind the choice of data will be clarified. The shift towards one main objective of understanding plume interannual variability, and the de-emphasis on validation should also help to clarify the choice of data.

3. The paper mentions a validation/intercomparison of satellite SSS products. To strengthen this aspect, I suggest including an example figure showing the different products for comparison. Additionally, information on the p-values of your correlations, bias, and spectral analysis to determine the effective resolution of the different products should be included.

If reference is made to all four products throughout the main body of text, a figure containing all four products will be included.

4. The decision to use the median of the products for analysis should be justified. It might be more appropriate to use the product that best aligns with in-situ information, has a higher spatiotemporal resolution, or realistically agrees with the expected dynamics of the area. If the median approach is retained, the reasoning behind this choice should be elaborated.

If the median product is still used, its use will be justified.

**Results:**

1. The results section lacks concrete analysis and tangible results to support the discussed relationships with the Arctic Oscillation Index, and river runoff.

Further analysis on the AOI and lagged correlations with river runoff will be conducted.

**Discussion:**

1. The discussion/conclusion emphasizes that wind is the dominant driver of offshore or onshore Lena River plume transport. To strengthen this claim, the study should include additional evidence from the analysis correlating composites and other drivers, for example, the river runoff, ice melting, etc.

A lagged correlation analysis will be conducted between eastward wind stress, river runoff, SSS, SST and sea ice concentration. This will help to justify that runoff is not a dominant driver of variability in SSS.

2. Line 415: your claim that because GLORYS12V1, which doesn't include interannually varying river runoff, replicates the SSS pattern well as compared to satellite SSS, suggests that variability in river runoff is not a significant contributor

to the interannual variability in GLORYS12V1 SSS. However, GLORYS12V1 utilizes in-situ SSS data, which is how it reproduces the SSS pattern. This does not negate the potential influence of river runoff on interannual SSS variability. The absence of river runoff does not imply that it has no impact on SSS dynamics. Moreover, the correlation between GLORYS12V1 and satellite SSS patterns does not necessarily indicate causation, river runoff could influence Laptev SSS variability, if you make this argument, the paper should conduct a more comprehensive analysis that explicitly investigates the impact of river runoff on interannual SSS variability.

Lagged correlations with river runoff will be conducted.

3. In Section 4.1, there is a discussion about correlating composites to river runoff, but no results are shown to support the argument. The analysis should be included to provide tangible evidence for the discussion. Additionally, the use of the BEC SSS should be addressed if you want to compare your results to the study of the product as in Umbert et al. 2021, who uses this product. The absence of BEC SSS figures and the correlation against river runoff data should be explained to ensure a coherent argument.

We will include a comparison figure of all four satellite products or at least ensure that the reasoning behind our choice of products is more clear earlier in the methods section. The expanded analysis of river runoff will be compared with reference to Umbert et al., 2021

*Figures:*

1. Figure 1, it is unclear why the wind over plots are not the mean for September, similar to the salinity over plots. Providing an explanation for this difference would enhance the figure's clarity and interpretation.

The choice of wind metric will be better explained in text. Results from the lagged temporal correlation analysis between eastward turbulent surface stress and salinity (see Figure 1 below) should help to explain this choice.

[Figure]

*Figure 1: Lagged temporal correlation between ERA5 eastward turbulent surface stress in April-September (4-9) and GLORYS September SSS*

2. In Figure 3, I strongly suggest to include the other two satellite SSS products

Including the other two products will be considered and balanced with wishing to not overcomplicate the figure. Including all four products in another figure (possibly figure 2) may help to illustrate why it is not beneficial to include all four products here.

**Minor comments:**

*Introduction*

I suggest including a reference to Umbert et al. 2021 as it also uses SMOS SSS to characterize the Lena River plume in the Laptev Sea, which could provide valuable context and potential links between the two studies.

Reference to the Umbert et al., 2021 paper will be made in the introduction as well as in the discussion.

*Methods*

In line 280, it is mentioned that the median product is generated using GLORYS12V1, LOCEAN SMOS, and both SMAP products. However, it seems there might be a discrepancy, as it was previously stated that there were four satellite products. This inconsistency needs clarification.

The median product is only calculated from the four satellite SSS products. It is then compared with GLORYS SSS. The wording in this line is unclear so will be altered for clarification.

*Results*

Section 3.3 is missing, but it is referred to in the text as "3.2 3". The authors should correct this discrepancy and make sure the section numbers are accurate.

The section will be correctly renamed to resolve this.

In Table 1, it is puzzling that the median product has more observations than any of the individual products. The authors should address this discrepancy and provide an explanation for the data variations to ensure the table's accuracy and consistency.

If we chose to keep the median, the reasoning behind this will be clarified.

---

## Author Response (AR1)

**REVIEW 1** :

*Hi Phoebe,*

*You're not going to like what comes next. Sorry.*

We thank the reviewer for their honest and constructive review and believe it has helped to notably strengthen this paper.

*Major comments*

*I. The objective of this manuscript is unclear*

*Although both the abstract and the introduction are very long, they are also unfocused, so it is unclear whether the aim of this article is to:*

- *combine satellite, reanalysis, and in-situ data to study the plume;*

- *demonstrate that satellite data can be used to look at sea surface salinity in the region;*

- *or determine which reanalysis product is most adapted for this study.*

The main objective of the paper: "combining satellite, reanalysis, and in-situ data to study the plume" has been highlighted more clearly at the end of the introduction. The introduction has also been shortened to help clarify the focus of the paper. In particular, the initial introduction section focusing on Arctic-wide processes, has been removed, and given the de-emphasis on satellite data, the section providing a broad overview of satellite SSS data has also been condensed.

The abstract has also been updated to reflect these changes and to better emphasize the main objective. The key points have also been re-arranged and altered to reflect this.

*These different objectives would then result in different structures of the manuscript, with the majority of the product comparisons and validations going to the appendix. It would also affect the time period considered. The choice of in-situ data would then be affected as well; see my next point.*

The components of the manuscript validating satellite data used with in-situ data have been moved to the appendix.

*II. The choice of data, especially their resolution, was surprising*

*The method section did not clarify much, so I am not sure which time period you worked with. That is, you mention that you use UDASH, but that stopped in 2015 and would not really help with SMAP. The manuscript needs an overarching table that says for all types of products over which time period and at which spatio-temporal resolution they are available (not the resolution at which you use them; their native one).*

An overarching table has now been added as requested for the satellite products to clarify all products used, their spatial resolutions and time-periods (new Table 1). All satellite products are now used/re-gridded onto a 0.25 degree grid for clarity and to aid comparison.

*Coupled with the fact that the objective is unclear, I won't be able to give you a clear direction. But if you want to validate SSS, I would have looked for underway CTD data rather than CTD casts. It will be at approx. 10 m depth, but on most vessels (and especially so on Oden, i.e. for the SWERUS data) the upper 10 m of the CTD casts cannot be used anyway. If you want the upper water column, there might be ITPs nearby, and there should be at least one mooring, but I'm not sure of their time coverage.*

The section detailing in-situ data used has been updated to more accurately reflect the range of in-situ data used for validation (not just CTD casts). The SWERUS (Oden) cruise CTD data is no longer used. NABOS underway data is used instead of CTD casts from the 2018 cruise. Table A1 (previously Table 1) has been updated to reflect these changes.

ITPs from the Mosaic cruise (from 2020) were considered for use but were >84N (and notably away from the shallow Laptev Shelf and the plume). The NABOS moorings (2013-2015, 2018-2021) were also considered for validation but are too far off the shelf and too deep (minimum depth 30m) for validation of the main plume.

*The other thing that really surprised me is that you want to investigate a plume dynamics, have a 3-day product available… and downgrade it to monthly resolution. And then regularly show results in the manuscript that you explain with the poor resolution of your monthly product. Use the 3-day version.*

*And regardless of what you do, you need to say why you do so. Maybe you had a perfectly valid reason for using CTD casts and downgrading everything to one month, but you did not write it so the reader cannot know.*

The shift to one main objective and the increased focus on reanalysis data will have helped to clarify the choice of data. In particular, much of the text describing comparison with in-situ data has been moved to the appendix, and therefore is a less central focus of the paper.

The reasoning behind the choice to use monthly data has also been clarified in the methods section: "Higher temporal resolution satellite products were considered for analysis but comparison with in-situ data suggested they did not notably improve correlations with in-situ data. Therefore, these results did not justify their use over monthly products."

*III. Causation is not shown*

*This is my main issue with your manuscript. You do not demonstrate causation. You produce two composites, and declare that the variable you composited against explains the differences.*

*Let's start with the definition of the composites. I agree that on Figure 2, the circulation is different. However on Figure 4, the uncertainty is so large that some of the strongest years could in fact have a value with either sign. See for example 2012 and 2019. As rotation is involved, a metric based on the curl of the wind, or simply on the sea level pressure, may be more effective and robust.*

To demonstrate causation, a lagged correlation analysis has now been run between ERA5 eastward turbulent surface stress in individual months and GLORYS SSS in September. The results of this analysis now establish that eastward wind stress is the dominant driver of September SSS.   The correlation is strongest in June, July and August so we now use a 3-month metric (June-August) (rather than the 4-month metric previously being used).

Note: In Figure 7 (previously Figure 4), the grey overlay is the maximum and minimum wind stress over the 4 month period (not the uncertainty). As eastward wind stress in September is highly variable, the range in surface stress over this three month period is still intentionally left on Figure 7 to help interpretation of individual years.

*Anyway, the outcome of the composites is that the SSS looks different. But so do the sea ice and the SST, which could both explain the SSS pattern, and even be responsible for the wind differences. Or maybe wind and SSS are both the result of another variable that is not included in your analysis.*

*What you are really showing is that the hypothesis that wind drives SSS is not incompatible with the observations. But as your analyses currently are, you do not demonstrate the causality. One option is therefore to just rephrase everything, removing all mentions of "the wind drives" and saying what I just wrote. But that's rather underwhelming a result.*

*Instead, you could utilise the 3-day product in its full 3-day glory. For each point, or for the overall region, do a lagged (temporal) correlation analysis of the relationship between wind (available at daily resolution; downgrade to 3-day) and the 3-day SSS from BEC. Then see which variable drives the other, based on these values. I personally would push the analysis and perform the same calculation also with the SST and the sea ice, and have an overall result matrix that shows for each pair of variables for which lag the correlation is maximal and what that correlation value is.*

A lagged-temporal correlation analysis has now been conducted of both wind stress and runoff with SSS using monthly rather than 3-daily data (see previous response on the 3 day product). A section has been added to the paper to discuss the results of this correlation analysis (depicted in Figures 4, 5 and 6). This shows that eastward turbulent surface stress in June-August is particularly strongly correlated with September SSS.

The same temporal correlation analysis was also conducted of both wind stress and runoff with SST and SIC. The results of this analysis have also been included in the paper. The differing spatial patterns highlighted by the correlation maps highlight the different processes controlling SSS and SST/SIC.

Whilst sea ice formation does play a strong role in controlling salinity in winter, sea ice melt (and precipitation) contribute several orders of magnitude too little freshwater to be dominant drivers of SSS variability in summer. The minimal direct role of sea ice melt has been clarified earlier in the text with reference to prior literature in the introduction.

*Less major comments, in order of appearance*

*Salinity unit: You meant "psu"or really"pss"? Oceanographers use the absolute salinity in g/kg now.*

We tried altering the paper to not use any salinity unit but found this to be unclear. Whilst the difference between absolute and practical salinity is negligible, especially compared to the uncertainty in salinity in this region, both satellite and GLORYS12V1 SSS are provided as practical salinities. Therefore, we choose to use units for practical salinity rather than absolute salinity. We avoided the use of "psu" following the guidelines of the TEOS manual "Note that Practical Salinity is a unit-less quantity. Though sometimes convenient, it is technically incorrect to quote Practical Salinity in "psu"; rather it should be quoted as a certain Practical Salinity "on the Practical Salinity Scale PSS-78"." (Intergovernmental Oceanographic Commission et al., 2015).

*Subsection 2.1.1: not detailed enough. Either there or in the introduction, you need to be more specific about which satellite measures at which band, has which footprint, repeat time, etc. Something similar to the first paragraph of section 2.1 of these people: https://tc.copernicus.org/articles/12/921/2018/tc-12-921-2018.pdf , but for all sensors (at least SMOS and SMAP, and then explanations about how the different products combine them).*

As noted previously, our main objective is to study the plume rather than validate the satellite SSS data. Having made this clear, we do not feel it is necessary to include specific details about the satellite measurements.

*Subsection 2.2.1: also not detailed enough; what do you mean by "correlation" and "RMSD"? I assume that you took all points available, regardless of location and time, and basically did a regression? Given that the plume is both time and space dependent, as shown on your figure, I would recommend you verify the temporal and spatial accuracy separately. You may need more points for this, agreed, but see Major comment II.*

The details of the methodology are now included in the appendix. By "correlation" and "RMSD" we mean the Pearson correlation coefficient and root mean square difference between all in situ data that has a corresponding satellite SSS value across the entire region and time period. The reviewer is correct that we took all points available irrespective of space and time. This has been more clearly explained within the appropriate section of the methodology in the appendix.

*Section 3.1: You do not show the RMSD. See my previous comment anyway, but that could be added to the table -which could be shortened once the manuscript is more focused, see Major comment I.*

RMSD values have now been included in Table A2 in the appendix.

*Sea ice: Line 377 onwards you give statistics of the sea ice area, without specifying over which region. Overall, and based on the figures shown in the manuscript, the area does not seem to matter as much as the southern / eastern extent. I would rather use such extent, if doing the correlations suggested above. If not, then do not even quantify it; your maps are very clear.*

The sea ice area metric is no longer included in the paper, as it was deemed to no longer be needed with the inclusion of the sea ice concentration correlation plots.

*Runoff: Line 405, the discussion starts with an analysis of the correlation between runoff and SSS. It is not specified, but I assume that the runoff values are published elsewhere? If so, reproduce them here, and do a proper (lagged) correlation analysis.*

Interannual variability in GRO runoff is now considered and correlated to variability in the spatial pattern of GLORYS12V1 SSS, SST and SIC. A section has been added to the paper to discuss the results of this correlation analysis (depicted in Figures 4, 5 and 6).

*Arctic Oscillation: Same comment, no information about where the AOI comes from and the correlation analysis is not shown.*

The source of Arctic Oscillation data has been clarified in the methods section.

*REVIEW 2* :

*Dear coauthors,*

*Your paper offers valuable insights into the drivers of Laptev Sea dynamics and interannual variability in salinity and temperature. It provides evidence that the salinity and temperature signatures agree in different reanalyses and satellite products under different wind regimes. Addressing the major comments proposed will improve the clarity and robustness of the research, enhancing its contribution to the scientific community.*

We thank you for your positive review and agree that implementing the suggested changes will strengthen the paper.

*Major Comments:*

*The focus of the paper:*

*The stated objective of the paper is to determine the drivers of the interannual variability of the Laptev Sea dynamics. However, a significant portion of the paper is devoted to the validation/intercomparison of satellite Sea Surface Salinity (SSS) products. While the effort to validate and compare these products is commendable, it appears to dominate the narrative, diverting attention from the primary objective of identifying the drivers of Laptev Sea dynamics.*

*To address this concern, I see two options:*

*a) Dividing the current study into two separate papers, each with a distinct focus and stating clearly the objectives (Validation and Intercomparison of Satellite SSS Products / Drivers of Laptev Sea Interannual Variability in Salinity and Temperature).*

*b) Focus on the primary objective during the narrative and if you feel that the validation/intercomparison of satellite SSS products is an essential part of the methodology, it would be beneficial to include at least one example figure showcasing the different products. Additionally, providing information on the p-value of your correlations, bias, and spectral analysis to assess the effective resolution of the products will improve the methodological rigor and transparency of the study.*

We agree that the validation distracts from the main objective. The abstract and introduction have been shortened to help more clearly highlight the focus of the paper. The section validating in-situ data has been moved to the appendix to help better emphasize the main objective.

**Methods:**

1. *In subsection 2.2.2, it is not clear why the analysis is not performed with all four satellite products. To ensure the robustness of the study, it would be beneficial to explain the reasons for the exclusion, if any, of certain products in the analysis.*

All four products have now been included in Figure 2. As the validation section has been moved out of the results section and into the appendix, discussion of correlation and RMSD values now occurs alongside this figure which helps to clarify the choice to exclude BEC and RSS.

2. *The temporal resolution of satellite SSS products is a critical factor when studying the dynamics of a region like the Laptev Sea, where rapid changes can occur over short time scales. If you choose not to use 3-day or available 8-day satellite Sea Surface Salinity (SSS) products, providing a clear and well-justified argument for this decision is crucial.*

The reasoning behind the choice of data has been clarified in the methods section: "Higher temporal resolution satellite products were considered for analysis but comparison with in-situ data suggested they did not notably improve correlations with in-situ data. Therefore, these results did not justify their use over monthly products.

3. *The paper mentions a validation/intercomparison of satellite SSS products. To strengthen this aspect, I suggest including an example figure showing the different products for comparison. Additionally, information on the p-values of your correlations, bias, and spectral analysis to determine the effective resolution of the different products should be included.*

All four products have now been included in Figure 2. P values for each correlation have not been included in Table A1 as they are all << 0.01 but the title has been updated to reflect this.

4. *The decision to use the median of the products for analysis should be justified. It might be more appropriate to use the product that best aligns with in-situ information, has a higher spatiotemporal resolution, or realistically agrees with the expected dynamics of the area. If the median approach is retained, the reasoning behind this choice should be elaborated.*

The median product is no longer used in analysis as we feel it is more useful to include each of the four products to more clearly illustrate their similarities/differences.

**Results:**

1. *The results section lacks concrete analysis and tangible results to support the discussed relationships with the Arctic Oscillation Index, and river runoff.*

Lagged correlation analysis has been conducted with GRO runoff and SSS, SST and SIC and has been included as an additional results section. A section has been added to the paper to discuss the results of this correlation analysis (depicted in Figures 4, 5 and 6).

The AOI timeseries has been added to the eastward turbulent surface stress timeseries to enable visual comparison and better support the discussion in text.

**Discussion:**

1. *The discussion/conclusion emphasizes that wind is the dominant driver of offshore or onshore Lena River plume transport. To strengthen this claim, the study should include additional evidence from the analysis correlating composites and other drivers, for example, the river runoff, ice melting, etc.*

A lagged correlation analysis has been conducted between eastward wind stress, river runoff, and SSS, SST and SIC. The discussion section has been updated to discuss the results of this correlation analysis (depicted in Figures 4, 5 and 6).

2. *Line 415: your claim that because GLORYS12V1, which doesn't include interannually varying river runoff, replicates the SSS pattern well as compared to satellite SSS, suggests that variability in river runoff is not a significant contributor to the interannual variability in GLORYS12V1 SSS. However, GLORYS12V1 utilizes in-situ SSS data, which is how it reproduces the SSS pattern. This does not negate the potential influence of river runoff on interannual SSS variability. The absence of river runoff does not imply that it has no impact on SSS dynamics. Moreover, the correlation between GLORYS12V1 and satellite SSS patterns does not necessarily indicate causation, river runoff could influence Laptev SSS variability, if you make this argument, the paper should conduct a more comprehensive analysis that explicitly investigates the impact of river runoff on interannual SSS variability.*

The section described has now been altered to no longer make this claim. This section has also been expanded to include the results of the lagged correlation analysis.

3. *In Section 4.1, there is a discussion about correlating composites to river runoff, but no results are shown to support the argument. The analysis should be included to provide tangible evidence for the discussion. Additionally, the use of the BEC SSS should be addressed if you want to compare your results to the study of the product as in Umbert et al. 2021, who uses this product. The absence of BEC SSS figures and the correlation against river runoff data should be explained to ensure a coherent argument.*

Correlations with river runoff have now been included in results and discussion sections.

Figure 2 now includes a comparison of all four satellite products which helps to explain the reasoning behind the choice to exclude BEC. The choice to exclude BEC from analysis makes comparison with the Umbert et al. 2021 paper more challenging. However, the discussion and comparison with Umbert et al. (2021) has been expanded upon based on the lagged-correlation analysis between runoff and GLORYS SSS variability.

***Figures:***

1. *Figure 1, it is unclear why the wind over plots are not the mean for September, similar to the salinity over plots. Providing an explanation for this difference would enhance the figure's clarity and interpretation.*

The choice of wind metric was derived from the lagged-correlation analysis between wind stress and September SSS. June, July and August were found to have the strongest correlations with September SSS so a mean of these months was chosen as the wind metric. The strong correlations with June-August can be seen in Figure 7. This has also been more clearly explained in text in the methods section.

2. *In Figure 3, I strongly suggest to include the other two satellite SSS products*

All four products have now been included in Figure 2.

**Minor comments:**

**Introduction**

*I suggest including a reference to Umbert et al. 2021 as it also uses SMOS SSS to characterize the Lena River plume in the Laptev Sea, which could provide valuable context and potential links between the two studies.*

Reference to the Umbert et al., 2021 paper has now been made in the introduction as well as in the discussion. The comparison in the discussion section has also been expanded to incorporate correlation results.

**Methods**

*In line 280, it is mentioned that the median product is generated using GLORYS12V1, LOCEAN SMOS, and both SMAP products. However, it seems there might be a discrepancy, as it was previously stated that there were four satellite products. This inconsistency needs clarification.*

The median product is no longer used. This sentence has been changed to reflect this.

**Results**

*Section 3.3 is missing, but it is referred to in the text as "3.2 3". The authors should correct this discrepancy and make sure the section numbers are accurate.*

The section has been correctly renamed to resolve this.

*In Table 1, it is puzzling that the median product has more observations than any of the individual products. The authors should address this discrepancy and provide an explanation for the data variations to ensure the table's accuracy and consistency.*

The median product is no longer included in the paper so this discrepancy is no longer present.

---

## Author Response (AR2)

*From the response to reviewers, it looked like all my comments had been carefully addressed. This was not the case in the manuscript, which, besides, had many formatting and other obvious errors, leading me to wonder whether I received the correct version. Feel free to ignore some of my comments below if indeed, this is simply an upload/version error.*

Apologies, indeed the version submitted was missing multiple applied formatting changes and typo corrections that were lost in version control.

*Major comments*

*Section 2 is still not detailed enough, which makes the entire paper unclear. That is, one cannot reproduce the analyses with the few information provided in section 2. For example, I had to wait until section 4 to understand that the correlation analyses had been performed with the complete time series of GLORYS12V1, not just the 10 years that overlap with SMOS/SMAP – the only time series shown. And I am still unsure whether the correlation AOI – wind stress in section 3.3 was performed on the only 12 points shown or on a longer series.*

*Solutions: Have all the products in table 1, not just the SSS ones, and make table A1 more detailed. In fact, alongside table A1, consider having a supplementary map or showing the most relevant cruise tracks on Fig 1. Then, for every single figure, table, and result described, say very clearly which time period you are using in the caption / in the panel titles.*

The time period over which the lagged correlation analyses are calculated (the full GLORYS timeseries) has been clarified in the methods section, when the lagged correlation analysis is first mentioned.

The time period used to calculate correlations between spring runoff, eastward turbulent surface stress and the AOI has also been clarified in the methods sections. Previously, the correlation coefficients quoted were calculated over the satellite (2010-2022) timeseries. They are now calculated over both the satellite timeseries (2010-2022) and over a longer timeseries (1993-2022).

Where they were not already present, the time periods relevant to each figure have been added in the associated figure caption. When discussing results relevant to a certain time period in text, the relevant time period used has been re-stated.

A table detailing all the reanalysis products used has been added to the methods section detailing reanalysis products.

A map depicting the in-situ data used for validation of satellite and reanalysis products has been included in the appendix.

*Also, now that you show more clearly the performances of the various reanalyses: Why GLORYS12V1? In tables A2 and A3, GloSea5 and ORAS5 clearly outperform it over the two satellite time periods.*

GLORYS12V1 was primarily chosen as it has been widely used in previous studies and hence its properties are well established in the Arctic and North Atlantic (Lellouche *et al.*, 2021; Verezemskaya *et al.*, 2021; Huang *et al.*, 2023).

One of the primary drivers of comparison with reanalysis was to be able to investigate stratification dynamics to compliment the view of the surface obtained from satellite data. GLORYS12V1 has previously been used for mixed layer studies (Hordoir *et al.*, 2022) and was the only model

considered which regularly had mixed layer depths <10m (and therefore was not entirely well-mixed everywhere on the shelf).

*Section 3.2 is better (even though that is not what I meant by lagged correlation, but never mind). The typos in the figure captions confused me for a moment: Figure 5, second sentence should read SST, not SSS (same on A2, appendix B); Figure 6, SIC, not SST. And latest in that section, but ideally already in section 3.1, you ought to show GLORYS's SST. You describe it a lot, use it regularly, even have section 3.4 dedicated to it – show, similar to Figure 3, how it performs in 3D.*

These typos have now been corrected.

A similar figure to Figure 3 for SST has been added in the appendix. A more in depth analysis of SST is not done as SST is highly variable in this region at this time of year (of over 5 degrees between September and October). Therefore, as is mentioned in text "SST, and in turn stratification in temperature also vary considerably over the course of September, so a higher temporal resolution analysis would be needed for investigating temperature stratification dynamics."

*One last analysis comment: Especially relevant based on what you discuss in section 4.4, why do you not perform any water mass analysis? Or at least properly verify the correlation between SST and SSS instead of just discussing similarities in the maps? This is the Ocean Science journal after all: Show us a T-S diagram, colour-coded by years.*

Whilst some of this analysis has been conducted (correlations between SSS and SST were calculated and a T-S diagram was plotted), given the complexity of the relationship between SSS and SST in this region, further analysis is needed to interpret these, which is beyond the scope of this paper.

*Minor comments*

*Overall, the paper remains too long and repetitive. With the exception of section 2 (see major comment), every sentence should be examined, whether it brings anything new and important to the story assessed, and if necessary be deleted. You could start with the undeserved superlatives, such as:*

*Line 245: 0.79 is not "notably" lower than 0.86, it is lower.*

*Lines 480 and 482, you do not provide a "complete" analysis; you are looking at the interannual variability of the monthly September values over a short time period.*

*Line 504: 0.49 is not "strongly" correlated.*

*Line 553: the gradient is not "clearly" visible (since you do not show the SST).*

*Line 616: the difference is not "considerable", at least not in the common usage of the term.*

The superlatives directly mentioned (and a number of others) have been removed. Several sentences have been shortened / removed to try and decrease repetition and help shorten the paper, whilst still ensuring the clarifications raised in review are also accommodated.

*Other minor comments, in order of appearance:*

*Line 106: Your introduction remains long and a bit unfocused. Help the reader by stating here "In this manuscript, we [objectives]"*

This phrasing has been added where the objectives are detailed, at the end of the introduction.

*Line 124: Since you already refer to the in-situ data there, consider swapping the subsections and starting with 2.1.3.*

The data products section now begins by describing the in-situ data, then the reanalysis data then the satellite data.

*Line 126: As it currently reads, it still feels like one of the objectives of the manuscript is to determine which reanalysis is best. Clarify this better, and refer to the appendix.*

This line has been altered to reflect the aim of comparing in-situ and satellite SSS with reanalyses, rather than a reanalysis intercomparison.

*Line 147 and throughout the manuscript (the usage of pss): I see your response, and would like to redirect you to the instructions you say you are following. Put more clearly: When working with practical salinity, option 1 (less correct) is to write psu as a unit; option 2 (most correct but rarely done) is to say at the very beginning of the manuscript that "salinities are quoted on the practical salinity scale" and then not write any unit. So on line 147 this would read as "SSS uncertainty is lower than 1".*

As per the response to both reviews, "We tried altering the paper to not use any salinity unit but found this to be unclear. Whilst the difference between absolute and practical salinity is negligible, especially compared to the uncertainty in salinity in this region, both satellite and GLORYS12V1 SSS are provided as practical salinities. Therefore, we choose to use units for practical salinity rather than absolute salinity. We avoided the use of "psu" following the guidelines of the TEOS manual "Note that Practical Salinity is a unit-less quantity. Though sometimes convenient, it is technically incorrect to quote Practical Salinity in "psu"; rather it should be quoted as a certain Practical Salinity "on the Practical Salinity Scale PSS-78"." (Intergovernmental Oceanographic Commission et al., 2015)."

*Line 228: "anomalous" is the wrong word. They would be anomalous if they were both compared to the mean of figures A3+A4… which maybe is what you mean, but in this case you have to say it and say over which time period considered. I rather suspect you mean that they are clearly different from each other.*

The wording has been changed to "notably different" rather than "anomalous" to reflect that this sentence is comparing the two years from each other rather than differences from a climatology.

*Line 230: "interannual variability" is not the correct word – again, I suspect you simply mean differences, unless you meant to add a reference to figures A3 and A4.*

The wording here has been changed.

*Line 237-241: Same comment as on the previous version. You have a 3-day product. You can verify this hypothesis for the satellites vs in-situ data.*

The wording of this line has been changed to reflect that satellite data clearly aligns more closely with in-situ data when using a daily satellite product for the relevant day. This is very clear in 2019 (where there is sufficiently abundant in-situ data which nicely captures the entire salinity front) and visible, but less clear, in 2016.

*Line 245 and Table A2: There are some very high correlations in there. Did you verify that not outlier is skewing the correlations?*

Yes, scatterplots were generated to ensure correlations are not influenced by the presence of outliers. In the case of BEC, the correlation deteriorates for values <25 pss.

**Line 255-259: Add a reference to the sea ice contours shown on Fig 2.**

A reference to Figure 2 has been included here.

**Line 290: "bottom waters", wrong word. Bottom waters are a specific water mass, found in the Arctic deeper than 2500 m. Here you mean "fresher water, below the surface", or "in the subsurface".**

This has been changed to "fresher subsurface waters."

**Line 313: "Vilkitsky Strait"; mark this location on Fig 1.**

The location of the Vilkiysky Strait has been added to Figure 1.

**Figure 7 and line 380: Same comment as on the previous version, even though you said you changed this in the response. With the uncertainty, 2012 could be in either direction. What is the impact on the results of not including that year?**

The JPL composite does not include 2012 and gives similar results (but does include 2021). If both these years (2012 and 2021) are excluded from the eastward composite, the SSS pattern remains very similar. The composite of just 2016 and 2017 is almost identical in the Laptev Sea (to the 3-year composites) but the low salinity signal that extends around the coast in the East Siberian Sea is stronger in the 2016-2017 composite, as it is particularly strong in 2016.

As per the response to both reviews "Note: In Figure 7 (previously Figure 4), the grey overlay is the maximum and minimum wind stress over the 4 month period (not the uncertainty)."

**Figure 7: The AOI standardised by the standard deviation of the surface stress is a strange measure. Why not have a second y-axis to the right, or even using the same axis as the runoff? The values should be in the same range (-4 to 4)**

A second y-axis has been added to this figure for the AOI.

**Lines 419-421: Same comment as on the previous version. These area values are meaningless without more clearly defining the regions, and are anyway not useful for the analysis. Remove this sentence.**

These area values have now been removed.

**Line 473: You should show the full runoff timeseries somewhere, so that the reader can see whether the satellite period is anomalous.**

The satellite period is not particularly anomalous (Figure 1). This has now been stated in text in the results section, where the shorter spring runoff timeseries is presented.

[Figure]

*Figure 1: Timeseries of Lena river runoff over 1993-2022*

Whilst we chose to not show the full runoff timeseries in text, to help not lengthen the paper further, GLORYS12V1 correlation coefficients are all calculated over 1993-2020. In addition, we now include correlation coefficients (between spring runoff and eastward turbulent surface stress and the AOI) over a longer 1993-2022 timeseries, as well as over the short 2010-2022 satellite timeseries. The relative consistency of correlation coefficients over these two time periods, helps to show that spring runoff in the satellite period is relatively consistent with earlier variability in spring runoff.

*Lines 493-495: Same comment as on the previous version. You can calculate the angle between the wind and the coast. You choose not to, for some reason. Ok, but in this case, you cannot write such a sentence.*

This sentence has been altered to not discuss the angle of plume transport from the wind.

*Lines 530-532: What do you mean? You have 30-years worth of data, you could verify this. Either don't write such a sentence, or say why you would not do the test.*

In the following paragraph, we explain that this hypothesis is not tested because "the constant well-mixed plume nearshore suggests GLORYS12V1 is not capable of fully representing plume stratification dynamics in this complex environment." In order to test this hypothesis, you would need a model that appears capable of accurately representing interannual variability in the mixed layer.

*Line 536: TOPAZ is the only one built on HYCOM; the other four are NEMO-based. It is not surprising that they suffer from the same bias when 4/5 are basically variations on the same model.*

The aim of this paper was not to do a reanalysis inter-comparison, so an extensive product comparison was not done and other products were not considered for use. It would be interesting to see how a wider array of models perform in this region (in particularly if any manage to capture both the surface interannual variability and the variability in stratification) but this is beyond the scope of this study.

*Line 541: Not fishing for references here, but yes, that is a known issue with models: The plume needs to move horizontally and vertically with this type of vertical grid (z-level), so even if it had perfectly accurate properties at the beginning (which it does not because of other biases), every time it gets to another grid cell it is strongly mixed with the ambient water, so you lose the signal quickly. Likewise, the issue of the too-well mixed shelf in z-level models has been reported on repeatedly. I would actually expect TOPAZ to perform better, since it is based on a sigma-level model, whereas GLORYS is z-level.*

This paragraph has now been altered to reflect that the z-level vertical grids in the models used here are a likely cause of the overmixing issue, based on inclusion of a relevant paper that was previously missed.

The initial plan was to use TOPAZ for this work but, at the time, the version available had only 28 levels, the shallowest of which was 5m (which was decided to be too deep for comparison with satellite SSS). Both this version and even in the more recent TOPAZ version with 40 levels, TOPAZ does not capture interannual variability of the surface plume visible in satellite or SSS data, and appears to also have an overmixing issue, so we chose to use GLORYS12V1.

*Line 588: Same comment as on the previous version, you are not showing the "initial" plume propagation. This would require that you use higher temporal resolution data.*

This word "initial" has been removed.

*Lines 590-592: Same comment as on the previous version, you cannot say who is a "dominant control" on what if you do a same-month correlation. Besides, you yourself right after seem to hint at the possibility that the correlation goes the other way round (less ice = more ocean exposed = warmer ocean).*

The beginning of this paragraph has been altered to clarify that although there is clearly strong correspondence between SST and SIC, neither can be assumed to be the dominant driver of the other given work shown here, and that it is likely that they both feedback on each other.

*Line 651: Which "increase in correlation strength over the more recent time period"? You either need a reference, or to remind the reader where you showed that (but you did not, right?).*

The increase in correlation strength was previously only discussed in the discussion section 4.2. These correlation coefficients have now been included in the results section, alongside the correlation coefficients for 2010-2022 (and the methods section has been updated to reflect this).

---

## Author Response (AR3)

**Dear authors,**

**thank you for the response to reviewers and the updated manuscript. I only have one last comment: in the response to the reviewers, you mention "correlations between SSS and SST were calculated and a T-S diagram was plotted" to analyse water masses present in the area, but such a diagram is not included in the manuscript. I also think that such a diagram, colour-coded by year for example, would help in improving our knowledge over this region, and I encourage you to include it in the manuscript.**

Thank you for your comments, we believe the addition of the T-S plot does strengthen the main findings.

A T-S plot, colour coded by years has been added to the appendix which includes correlation coefficients between SST and SSS in each panel (Figure A7). Given the reviewer suggestion that the paper is lengthy, this figure was not included in the main body of text to not further extend the paper. Sections 3.4 and 4.4 have been altered to reference and discuss the relevant findings from these T-S plots.